# BRIDGING DISCRETE AND CONTINUOUS RL: STABLE DETERMINISTIC POLICY GRADIENT WITH MARTINGALE CHARACTERIZATION

## ABSTRACT

The theory of discrete-time reinforcement learning (RL) has advanced rapidly over the past decades. Although primarily designed for discrete environments, many real-world RL applications are inherently continuous and complex. A major challenge in extending discrete-time algorithms to continuous-time settings is their sensitivity to time discretization, often leading to poor stability and slow convergence. In this paper, we investigate deterministic policy gradient methods for continuous-time RL. We derive a continuous-time policy gradient formula based on an analogue of the advantage function and establish its martingale characterization. This theoretical foundation leads to our proposed algorithm, CT-DDPG, which enables stable learning with deterministic policies in continuous-time environments. Numerical experiments show that the proposed CT-DDPG algorithm offers improved stability and faster convergence compared to existing discrete-time and continuous-time methods, across a wide range of control tasks with varying time discretizations and noise levels.

## 1 INTRODUCTION

Deep Reinforcement learning (RL) has achieved remarkable success over the past decade, powered by theoretical advances and the success of algorithms in discrete-time systems such as Atari, Go, and Large Language Models (Mnih et al., 2013; Silver et al., 2016; Guo et al., 2025). However, many real-world problems, such as robotic control, autonomous driving, and financial trading, are inherently continuous in time. In these domains, agents need to interact with the environment at an ultra-high frequency, underscoring the need for continuous-time RL approaches (Wang et al., 2020).

One major challenge in applying discrete-time RL to continuous-time environments is the sensitivity to the discretization step size. As the step size decreases, standard algorithms often degrade, resulting in exploding variance, poor stability, and slow convergence. While several works have attempted to resolve this issue with discretization-invariant algorithms (Tallec et al., 2019; Park et al., 2021), their underlying design principles are rooted in discrete-time RL. As a result, these methods are not robust when applied to *complex, stochastic*, and *continuous* real-world environments.

Recently there is a fast growing body of research on continuous-time RL (Yildiz et al., 2021; Jia & Zhou, 2022a;b; 2023; Zhao et al., 2023; Giegrich et al., 2024), including rigorous mathematical formulations and various algorithmic designs. However, most existing methods either rely on model-based assumptions, or consider stochastic policy, which is difficult to sample in continuous time, state and action spaces (Jia et al., 2025), and imposes Bellman equation constraints which are not feasible for implementation within deep RL frameworks. These challenges hinder the application of continuous-time RL framework in practice, leading to an important research question:

*Can we develop a theoretically grounded algorithm that achieves stability and efficiency for deep RL in continuous-time environments?*

In this paper, we address this question by investigating deterministic policy gradient (DPG) methods. We consider general continuous-time dynamics driven by a stochastic differential equation over a finite horizon. our main contributions are summarized as follows:

- In Sec. 3, we develop a rigorous mathematical framework for model-free DPG methods in continuous-time RL. Specifically, Thm. 3.1 derives the DPG formula based on the *advantage rate function*. Thm. 3.2 further utilizes a martingale criterion to characterize the advantage rate function, laying the foundation for subsequent algorithm design. We also provide detailed comparisons against existing continuous-time RL algorithms with stochastic policy and discuss their major flaws and impracticality in deep RL frameworks.

- In Sec. 4, we propose CT-DDPG, a novel and practical actor-critic algorithm with stability and efficiency in continuous-time environments. Notably, we utilize a multi-step TD objective, and prove its robustness to time discretization and stochastic noises in Sec. 4.2. For the first time, we provide the theoretical insights of the failure of standard discrete-time deep RL algorithms in continuous and stochastic settings.

- Through extensive experiments in Sec. 5, we verify that existing discrete/continuous time algorithms lack robustness to time discretization and dynamic noise, while our method exhibits consistently stable performance.

## 2   PROBLEM FORMULATION

This section formulates the continuous RL problem, where the agent learns an optimal parametrized policy to control an unknown continuous-time stochastic system to maximize a reward functional over a finite time horizon. The extension to the infinite time horizon is technically similar, and hence omitted to avoid repetition.

Let the state space be $\mathbb{R}^n$ and the action space be an open set $\mathcal{A} \subseteq \mathbb{R}^d$. For each non-anticipative $\mathcal{A}$-valued control (action) process $\boldsymbol{a} = (a_t)_{t \geq 0}$, consider the associated state process governed by the following dynamics:

$$\mathrm{d}X_t^{\boldsymbol{a}} = b(t, X_t^{\boldsymbol{a}}, a_t)\mathrm{d}t + \sigma(t, X_t^{\boldsymbol{a}}, a_t)\mathrm{d}W_t, \ t \in [0, T]; \quad X_0^{\boldsymbol{a}} = x_0 \sim \nu, \tag{2.1}$$

where $\nu$ is the initial distribution, $(W_t)_{t \geq 0}$ is an $m$-dimensional Brownian motion on a filtered probability space $(\Omega, \mathcal{F}, \mathbb{F} = (\mathcal{F}_t)_{t \geq 0}, \mathbb{P})$, and $b : [0, T] \times \mathbb{R}^n \times \mathcal{A} \to \mathbb{R}^n$, $\sigma : [0, T] \times \mathbb{R}^n \times \mathcal{A} \to \mathbb{R}^{n \times m}$ are continuous functions. The reward functional of $\boldsymbol{a}$ is given by

$$\mathbb{E}\left[\int_0^T e^{-\beta t} r(t, X_t^{\boldsymbol{a}}, a_t)\mathrm{d}t + e^{-\beta T} g(X_T^{\boldsymbol{a}})\right], \tag{2.2}$$

where $\beta \geq 0$ is a discount factor, and $r : [0, T] \times \mathbb{R}^n \times \mathcal{A} \to \mathbb{R}$ and $g : \mathbb{R}^n \to \mathbb{R}$ are continuous functions, representing the running and terminal rewards, respectively.

It is well-known that under mild regularity conditions, it suffices to optimize (2.2) over control processes generated by Markov policies (Kurtz & Stockbridge, 1998). Given a Markov policy $\mu : [0, T] \times \mathbb{R}^n \to \mathcal{A}$, the associated state process $(X_t^{\mu})_{t \geq 0}$ evolves according to the dynamics:

$$\mathrm{d}X_t^{\mu} = b(t, X_t^{\mu}, \mu(t, X_t^{\mu}))\mathrm{d}t + \sigma(t, X_t^{\mu}, \mu(t, X_t^{\mu}))\mathrm{d}W_t, \ t \in [0, T]; \quad X_0^{\mu} = x_0 \sim \nu. \tag{2.3}$$

The agent aims to maximize the following reward

$$\mathbb{E}\left[\int_0^T e^{-\beta t} r(t, X_t^{\mu}, \mu(t, X_t^{\mu}))\mathrm{d}t + e^{-\beta T} g(X_T^{\mu})\right] \tag{2.4}$$

over all admissible policies $\mu$. Importantly, the agent does not have access to the coefficients $b$, $\sigma$, $r$ and $g$. Instead, the agent directly interacts with Eq. (2.3) with different actions, and refines her strategy based on observed state and reward trajectories. We emphasize that in this paper, we directly optimize (2.4) over deterministic policies, which map the state space directly to the action space, rather than over stochastic policies as studied in Jia & Zhou (2022b; 2023); Zhao et al. (2023), which map the state space to probability measures over the action space (see Sec. 3.3).

To solve Eq. (2.4), a practical approach is to restrict the optimization problem over a sufficiently rich class of parameterized policies. More precisely, given a class of policies $\{\mu_{\phi} : [0, T] \times \mathbb{R}^n \to \mathcal{A} \mid \phi \in \mathbb{R}^k\}$ parameterized by $\phi$, we consider the following maximization problem:

$$\max_{\phi \in \mathbb{R}^k} J(\phi), \quad \text{with} \quad J(\phi) := \mathbb{E}\left[\int_0^T e^{-\beta t} r(t, X_t^{\phi}, \mu_{\phi}(t, X_t^{\phi}))\mathrm{d}t + e^{-\beta T} g(X_T^{\phi})\right], \tag{2.5}$$

where $X^\phi$ denotes the state process controlled by $\mu_\phi$. Throughout this paper, we assume the initial state distribution $\nu$ has a second moment, and impose the following regularity conditions on the policy class and model coefficients.

**Assumption 1.** *There exists $C \geq 0$ such that for all $t \in [0, T]$, $a, a' \in \mathcal{A}$ and and $x, x' \in \mathbb{R}^n$,*

$$|b(t, x, a) - b(t, x', a')| + |\sigma(t, x, a) - \sigma(t, x', a')| \leq C(|x - x'| + |a - a'|),$$
$$|b(t, 0, 0)| + |\sigma(t, 0, 0)| \leq C, \quad |r(t, x, a)| + |g(x)| \leq C(1 + |x|^2 + |a|^2),$$

*and there exists a locally bounded function $\rho_1 : [0, \infty) \to [0, \infty)$ such that for all $\phi \in \mathbb{R}^k$, $t \in [0, T]$, and $x, x' \in \mathbb{R}^n$, $|\mu_\phi(t, x) - \mu_\phi(t, x')| \leq \rho_1(|\phi|)|x - x'|$ and $|\mu_\phi(t, 0)| \leq \rho_1(|\phi|)$.*

Asp. 1 holds for all policies parameterized by feedforward neural networks with Lipschitz activations. It ensures that the state dynamics and the objective function are well defined for any $\phi \in \mathbb{R}^k$.

## 3 Main Theoretical Results

We will first characterize the gradient of the objective functional Eq. (2.5) with respect to the policy parameter $\phi$, using a continuous-time analogue of the discrete-time advantage function. We will then derive a martingale characterization of this continuous-time advantage function and value function, which serves as the foundation of our algorithm design under deterministic policies. All detailed proofs can be found in Appendix B.

### 3.1 Deterministic policy gradient (DPG) formula

We first introduce a dynamic version of the objective function $J(\phi)$. For each $(t, x) \in [0, T] \times \mathbb{R}^n$, define the value function

$$V^\phi(t, x) := \mathbb{E}\left[\int_t^T e^{-\beta(s-t)} r(s, X_s^\phi, \mu_\phi(s, X_s^\phi)) \mathrm{d}s + e^{-\beta(T-s)} g(X_T^\phi) \,\bigg|\, X_t^\phi = x\right]. \quad (3.1)$$

Note that $J(\phi) = \mathbb{E}_{x \sim \nu}[V^\phi(0, x)]$. We additionally impose the following differentiability condition on the model parameters and policies with respect to the parameter.

**Assumption 2.** *For all $(t, x) \in [0, T] \times \mathbb{R}^n$, $a \mapsto (b, \sigma\sigma^\top, r)(t, x, a)$ and $\phi \mapsto \mu_\phi(t, x)$ are continuously differentiable. There exists a locally bounded function $\rho_2 : [0, \infty) \to [0, \infty)$ such that for all $\phi \in \mathbb{R}^k$ and $(t, x) \in [0, T] \times \mathbb{R}^n$,*

$$\frac{|\partial_\phi b(t, x, \mu_\phi(t, x))|}{1 + |x|} + \frac{|\partial_\phi(\sigma\sigma^\top)(t, x, \mu_\phi(t, x))| + |\partial_\phi r(t, x, \mu_\phi(t, x))|}{1 + |x|^2} \leq \rho_2(|\phi|).$$

*Moreover, $V^\phi \in C^{1,2}([0, T] \times \mathbb{R}^n)$ for all $\phi \in \mathbb{R}^k$.*

Under Asp. 1, by Itô's formula, for any given $\phi \in \mathbb{R}^k$, $V^\phi \in C^{1,2}([0, T] \times \mathbb{R}^n)$ satisfies the following linear Bellman equation: for all $(t, x) \in [0, T] \times \mathbb{R}^n$,

$$\mathcal{L}[V^\phi](t, x, \mu_\phi(t, x)) + r(t, x, \mu_\phi(t, x)) = 0, \quad V^\phi(T, x) = g(x), \quad (3.2)$$

where $\mathcal{L}$ is the generator of (2.3) such that for all $\varphi \in C^{1,2}([0, T] \times \mathbb{R}^n)$,

$$\mathcal{L}[\varphi](t, x, a) := \partial_t \varphi(t, x) - \beta\varphi(t, x) + b(t, x, a)^\top \partial_x \varphi(t, x) + \frac{1}{2}\mathrm{Tr}(\Sigma(t, x, a)\partial_{xx}^2 \varphi(t, x)), \quad (3.3)$$

with $\Sigma := \sigma\sigma^\top$. The following theorem presents the DPG formula for the continuous RL problem.

**Theorem 3.1.** *Suppose Asps. 1 and 2 hold. For all $(t, x) \in [0, T] \times \mathbb{R}^n$ and $\phi \in \mathbb{R}^k$,*

$$\partial_\phi V^\phi(t, x) = \mathbb{E}\left[\int_t^T e^{-\beta(s-t)} \partial_\phi \mu_\phi(s, X_s^\phi)^\top \partial_a A^\phi(s, X_s^\phi, \mu_\phi(s, X_s^\phi)) \mathrm{d}s \,\bigg|\, X_t^\phi = x\right],$$

*where $A^\phi(t, x, a) := \mathcal{L}[V^\phi](t, x, a) + r(t, x, a)$.*

The proof of Thm. 3.1 follows by quantifying the difference between the value functions corresponding to two policies, and then applying Vitali's convergence theorem. Similar formula was established in Gobet & Munos (2005) under stronger conditions that the running reward $r$ is zero, the diffusion coefficient is uniformly elliptic, and the coefficients are four times continuously differentiable.

**Remark 1.** *Thm. 3.1 is analogous to the DPG formula for discrete-time Markov decision processes (Silver et al., 2014). The function $A^\phi$ plays the role of advantage function used in discrete-time DPG, and has been referred to as the advantage rate function in Zhao et al. (2023). To see it, assume $\beta = 0$, and for any given $N \in \mathbb{N}$, consider the discrete-time version of Eq. (2.5):*

$$J_{\Delta t}(\phi) := \mathbb{E}\big[\sum_{i=0}^{N-1} r(t_i, X_{t_i}^{\Delta t, \phi}, \mu_\phi(t_i, X_{t_i}^{\Delta t, \phi}))\Delta t + g(X_T^{\Delta t, \phi})\big], \tag{3.4}$$

*where $\Delta t = T/N$, $t_i = i\Delta t$, and $X^{\Delta t, \phi}$ satisfies the following time-discretization of Eq. (2.3):*

$$X_{t_{i+1}}^{\Delta t, \phi} = X_{t_i}^{\Delta t, \phi} + b(t_i, X_{t_i}^{\Delta t, \phi}, \mu_\phi(t_i, X_{t_i}^{\Delta t, \phi}))\Delta t + \sigma(t_i, X_{t_i}^{\Delta t, \phi}, \mu_\phi(t_i, X_{t_i}^{\Delta t, \phi}))\sqrt{\Delta t}\omega_{t_i},$$

*and $(\omega_{t_i})_{i=0}^{N-1}$ are independent standard normal random variables. By the deterministic policy gradient formula (Silver et al., 2014),*

$$\partial_\phi J_{\Delta t}(\phi) = \mathbb{E}\Big[\sum_{i=0}^{N-1} \partial_\phi \mu_\phi(t_i, X_{t_i}^{\Delta t, \phi})^\top \partial_a A^{\Delta t, \phi}(t_i, X_{t_i}^{\Delta t, \phi}, \mu_\phi(t_i, X_{t_i}^{\Delta t, \phi}))\Delta t\Big], \tag{3.5}$$

*where $A^{\Delta t, \phi}(t, x, a) := \dfrac{Q^{\Delta t, \phi}(t, x, a) - V^{\Delta t, \phi}(t, x)}{\Delta t}$ is the advantage function for Eq. (3.4) normalized with the time stepsize. As $N \to \infty$, $A^{\Delta t, \phi}$ converges to $A^\phi$, as shown in Jia & Zhou (2023). Sending $\Delta t \to 0$ in Eq. (3.5) yields the continuous-time DPG in Thm. 3.1.*

### 3.2 Martingale characterization of continuous-time advantage rate function

By Thm. 3.1, implementing the DPG requires computing the advantage rate function $A^\phi$ in a neighborhood of the policy $\mu_\phi$. The following theorem characterizes the advantage rate function through a martingale criterion.

**Theorem 3.2.** *Suppose Asps. 1 and 2 hold. Let $\phi \in \mathbb{R}^k$, $\hat{V} \in C^{1,2}([0, T] \times \mathbb{R}^n)$ and $\hat{q} \in C([0, T] \times \mathbb{R}^n \times \mathcal{A})$ satisfy the following conditions for all $(t, x) \in [0, T] \times \mathbb{R}^n$:*

$$\hat{V}(T, x) = g(x), \quad \hat{q}(t, x, \mu_\phi(t, x)) = 0, \tag{3.6}$$

*and there exists a neighborhood $\mathcal{O}_{\mu_\phi(t,x)} \subset \mathcal{A}$ of $\mu_\phi(t, x)$ such that for all $a \in \mathcal{O}_{\mu_\phi(t,x)}$,*

$$\left(e^{-\beta(s-t)}\hat{V}(s, X_s^{t,x,a}) + \int_t^s e^{-\beta(u-t)}(r - \hat{q})(u, X_u^{t,x,a}, \alpha_u)\mathrm{d}u\right)_{s \in [t, T]} \tag{3.7}$$

*is an $\mathbb{F}$-martingale, where $X^{t,x,a}$ satisfies for all $s \in [t, T]$,*

$$\mathrm{d}X_s^{t,x,a} = b(s, X_s^{t,x,a}, \alpha_s)\mathrm{d}s + \sigma(s, X_s^{t,x,a}, \alpha_s)\mathrm{d}W_s, \quad X_t^{t,x,a} = x, \tag{3.8}$$

*and $(\alpha_s)_{s \geq t}$ is a square-integrable $\mathcal{A}$-valued adapted process with $\lim_{s \searrow t} \alpha_s = a$ almost surely. Then $\hat{V}(t, x) = V^\phi(t, x)$ and $\hat{q}(t, x, a) = A^\phi(t, x, a)$ for all $(t, x, a) \in [0, T] \times \mathbb{R}^n \times \mathcal{O}_{\mu_\phi(t,x)}$.*

Thm. 3.2 establishes sufficient conditions ensuring that the functions $\hat{V}$ and $\hat{q}$ coincide with the value function and the advantage rate function of a given policy $\mu_\phi$, respectively. Eq. (3.6) requires that $\hat{V}$ agrees with the terminal condition $h$ at time $T$, and the function $\hat{q}$ satisfies the linear Bellman equation Eq. (3.2) as the true advantage rate $A^\phi$. The martingale constraint Eq. (3.7) ensures $\hat{q}$ is the advantage rate function associated with $\hat{V}$, for all actions in a neighborhood of the policy $\mu_\phi$.

To ensure exploration of the action space, Thm. 3.2 requires that the martingale condition Eq. (3.7) holds for state processes initialized with any action $a \in \mathcal{O}_{\mu_\phi(t,x)}$. In practice, one can use an exploration policy to generate these exploratory actions, which are then employed to learn the gradient of the target deterministic policy. This parallels the central role of off-policy algorithms in discrete-time DPG methods (Lillicrap et al., 2015; Haarnoja et al., 2018a).

### 3.3 IMPROVED EFFICIENCY AND STABILITY OF DETERMINISTIC POLICIES OVER STOCHASTIC POLICIES

Thm. 3.2 implies that DPG can be estimated both more efficiently and more stably than stochastic policy gradients, since it avoids costly integrations over the action space.

Recall that Jia & Zhou (2022b; 2023); Zhao et al. (2023) study continuous-time RL with stochastic policies $\pi : [0, T] \times \mathbb{R}^d \to \mathcal{P}(\mathcal{A})$ and establish an analogous policy gradient formula based on the corresponding advantage rate function. By incorporating an additional entropy term into the objective, Jia & Zhou (2023) characterizes the advantage rate function analogously to Thm. 3.2, replacing the Bellman condition Eq. (3.6) with

$$\mathbb{E}_{a \sim \pi(\mathrm{d}a|t,x)}[\hat{q}(t,x,a) - \gamma \log \pi(a|t,x)] = 0, \quad \forall (t,x) \in [0,T] \times \mathbb{R}^n, \tag{3.9}$$

where $\gamma > 0$ is the entropy regularization coefficient, and requiring the martingale constraint Eq. (3.7) to hold for all state dynamics starting at state $x$ at time $t$, with actions sampled randomly from $\pi$ at any time partition of $[0, T]$. Implementing the criterion Eq. (3.9) requires sampling random actions from the policy $\pi$ to compute the expectation over the action space. This makes policy evaluation substantially more challenging in deep RL, particularly with high-dimensional action spaces or non-Gaussian policies, often resulting in training instability and slow convergence, as observed in our experiments in Sec. 5. In contrast, the Bellman condition Eq. (3.6) for DPG can be straightforwardly implemented using a simple re-parameterization (see Eq. (4.1)).

## 4 ALGORITHM AND ANALYSIS

### 4.1 ALGORITHM DESIGN

Given the martingale characterization (Thm. 3.2), we now discuss the implementation details in a continuous-time RL framework via deep neural networks. We use $V_\theta, q_\psi, \mu_\phi$ to denote the neural networks for value, advantage rate function and policy, respectively.

**Martingale loss.** To ensure the martingale condition Eq. (3.7), let $M_t = e^{-\beta t} V_\theta(t, x_t) + \int_0^t e^{-\beta s}[r(s, x_s, a_s) - q_\psi(s, x_s, a_s)]\mathrm{d}s$. We adopt the following martingale orthogonality conditions (also known as generalized moment method) $\mathbb{E}\left[\int_0^T \zeta_t \mathrm{d}M_t\right] = 0$, where $\boldsymbol{\zeta} = (\zeta_t)_{[0,T]}$ is any test function. This is both necessary and sufficient to ensure the martingale condition for all $\mathbb{F}$-adapted and square-integrable processes $\boldsymbol{\zeta}$ (Jia & Zhou, 2022a; Bo et al., 2025).

In theory, one should consider all possible test functions, which leads to infinitely many equations. For practical implementation, however, it suffices to select a finite number of test functions with special structures. A natural choice is to set $\zeta_t = \partial_\theta V_\theta(t, x_t)$ or $\zeta_t = \partial_\psi q_\psi(t, x_t, a_t)$, in which case the marginal orthogonality condition becomes a vector-valued condition. The classic stochastic approximation method (Robbins & Monro, 1951) can be applied to solve the equation:

$$\theta \leftarrow \theta - \eta \partial_\theta V_\theta(t, x_t) \cdot \left(V_\theta(t, x_t) - \int_t^{t+\delta} e^{-\beta(s-t)}[r(s, x_s, a_s) - q_\psi(s, x_s, a_s)]\mathrm{d}s - e^{-\beta\delta} V_\theta(t+\delta, x_{t+\delta})\right),$$

$$\psi \leftarrow \psi - \eta \partial_\psi q_\psi(t, x_t, a_t) \cdot \left(V_\theta(t, x_t) - \int_t^{t+\delta} e^{-\beta(s-t)}[r(s, x_s, a_s) - q_\psi(s, x_s, a_s)]\mathrm{d}s - e^{-\beta\delta} V_\theta(t+\delta, x_{t+\delta})\right),$$

where $\delta > 0$ is the integral interval and the trajectory is sampled from collected data. Note that the update formula above is also referred as semi-gradient TD method in RL (Sutton et al., 1998).

**Bellman constraints.** To enforce Eq. (3.6), we re-parameterize the advantage rate function as

$$q_\psi(t, x, a) := \bar{q}_\psi(t, x, a) - \bar{q}_\psi(t, x, \mu_\phi(t, x)), \tag{4.1}$$

where $\bar{q}_\psi$ is a neural network and $\mu_\phi$ denotes the current deterministic policy (Tallec et al., 2019).

In practice, it is often challenging to design a neural network structure that directly enforces the terminal value constraint. To address this, we add a penalty term of the form: $\mathbb{E}(V_\theta(T, x_T) - g(x_T))^2$, where $x_T, g(x_T)$ are sampled from collected trajectories.

**Implementation with discretization.** Let $h$ denote the discretization step size. We denote by $\tilde{x}_t$ the concatenation of time and state $(t, x_t)$ for compactness. The full procedure of **C**ontinuous **T**ime **D**eep **D**eterministic **P**olicy **G**radient (CT-DDPG) is summarized in Alg. 1.

We employ several training techniques widely used in modern deep RL algorithms such as DDPG and SAC. In particular, we employ a target value network $V_{\theta^{tgt}}$, defined as the exponentially moving average of the value network weights. This technique has been shown to improve training stability in deep RL[1]. We further adopt a replay buffer to store transitions in order to improve sample efficiency. For exploration, we add independent Gaussian noises to the deterministic policy $\mu_\phi$.

**Multi-step TD.** When training advantage-rate net and value net, we adopt multiple steps $L > 1$ to compute the temporal difference error (see Eq. (4.2)). This is different from most off-policy algorithms which typically rely on a single transition step. Notably, when $L = 1$, our algorithm reduces to DAU (Tallec et al., 2019, Alg. 2) except that their policy learning rate vanishes as $h \to 0$. We highlight that multi-step TD is essential for the empirical success of CT-DDPG. In the next subsection, we theoretically demonstrate that one-step TD inevitably leads to gradient variance blow-up in the limit of vanishing discretization step, thereby slowing convergence.

## 4.2 ISSUES OF ONE-STEP TD IN CONTINUOUS TIME: VARIANCE BLOW UP

When training the value function $V_\theta$ and the advantage function $A_\psi$ for a given policy (stochastic or deterministic), *Temporal Difference* algorithms (Haarnoja et al., 2018a; Tallec et al., 2019; Jia & Zhou, 2023) typically use a one-step semi-gradient:

$$G_{\theta,h} := \frac{1}{h}\mathbb{E}\left[\partial_\theta V_\theta(\tilde{x}_t)\left(V_\theta(\tilde{x}_t) - (r_t - A_\psi(\tilde{x}_t, a_t)) \cdot h - e^{-\beta h}V_\theta(\tilde{x}_{t+h})\right)\right],$$
$$G_{\psi,h} := \frac{1}{h}\mathbb{E}\left[\partial_\psi A_\psi(\tilde{x}_t, a_t)\left(V_\theta(\tilde{x}_t) - (r_t - A_\psi(\tilde{x}_t, a_t)) \cdot h - e^{-\beta h}V_\theta(\tilde{x}_{t+h})\right)\right], \tag{4.3}$$

where $t \sim TruncExp(\beta; T)$ and $x_t \sim X_t^{\pi'}, a_t \sim \pi'(\cdot|t, x_t)$ with an exploration policy $\pi'$. In practice, however, one has to use stochastic gradient:

$$g_{\theta,h} := \frac{1}{h}\left[\partial_\theta V_\theta(\tilde{x}_t)\left(V_\theta(\tilde{x}_t) - (r_t - A_\psi(\tilde{x}_t, a_t)) \cdot h - e^{-\beta h}V_\theta(\tilde{x}_{t+h})\right)\right].$$
$$g_{\psi,h} := \frac{1}{h}\left[\partial_\psi A_\psi(\tilde{x}_t, a_t)\left(V_\theta(\tilde{x}_t) - (r_t - A_\psi(\tilde{x}_t, a_t)) \cdot h - e^{-\beta h}V_\theta(\tilde{x}_{t+h})\right)\right]. \tag{4.4}$$

**Proposition 4.1.** *Assume $\partial_\theta V_\theta, \|\partial_x V_\theta\|_{\sigma\sigma^\top}$ are not identically zero. Then the variance of stochastic gradient estimator blows up in the sense that:*

$$\lim_{h\to 0}\mathbb{E}[g_{\theta,h}] = \lim_{h\to 0}G_{\theta,h} = \Theta(1), \lim_{h\to 0}\mathbb{E}[g_{\psi,h}] = \lim_{h\to 0}G_{\psi,h} = \Theta(1), \tag{4.5}$$

$$\lim_{h\to 0} h \cdot \mathrm{Var}(g_{\theta,h}) = \Theta(1), \lim_{h\to 0} h \cdot \mathrm{Var}(g_{\psi,h}) = \Theta(1). \tag{4.6}$$

In contrast, Alg. 1 utilizes $L$-step TD loss with (stochastic) semi-gradient (for simplicity of the theoretical analysis, we consider hard update of target, i.e., $\tau = 1$):

$$G_{\theta,h,L} = \mathbb{E}\left[\partial_\theta V_\theta(\tilde{x}_t)\left(V_\theta(\tilde{x}_t) - \sum_{l=0}^{L-1} e^{-\beta lh}[r_{t+lh} - q_\psi(\tilde{x}_{t+lh}, a_{t+lh})]h - e^{-\beta Lh}V_\theta(\tilde{x}_{t+Lh})\right)\right], \tag{4.7}$$

$$g_{\theta,h,L} = \partial_\theta V_\theta(\tilde{x}_t)\left(V_\theta(\tilde{x}_t) - \sum_{l=0}^{L-1} e^{-\beta lh}[r_{t+lh} - q_\psi(\tilde{x}_{t+lh}, a_{t+lh})]h - e^{-\beta Lh}V_\theta(\tilde{x}_{t+Lh})\right). \tag{4.8}$$

**Proposition 4.2.** *Under the same assumptions in Prop. 4.1, if $Lh \equiv \delta > 0$, then the expected gradient does not vanish in the sense that*

$$\lim_{h\to 0}\mathbb{E}[g_{\theta,h,\frac{\delta}{h}}] = \lim_{h\to 0}G_{\theta,h,\frac{\delta}{h}} = \Theta(1). \tag{4.9}$$

*In addition, the variance of stochastic gradient does not blow up: $\overline{\lim}_{h\to 0}\mathrm{Var}(g_{\theta,h,\frac{\delta}{h}}) = \mathcal{O}(1)$.*

---

[1]Here we focus on a single target value network as our primary goal is to study the efficiency of deterministic policies in continuous-time RL. Extensions with multiple target networks (Fujimoto et al., 2018) can be readily incorporated.

---

**Algorithm 1** **C**ontinuous **T**ime **D**eep **D**eterministic **P**olicy **G**radient

---

**Inputs:** Discretization step size $h$, horizon $K = T/h$, discount rate $\beta$, number of episodes $N$, policy net $\mu_\phi$, advantage-rate net $\bar{q}_\psi$, value net $V_\theta$, update frequency $m$, trajectory length $L$, exploration noise $\sigma_{\text{explore}}$, soft update parameter $\tau$, learning rate $\eta$, batch size $B$, terminal value constraint weight $\alpha$

**Learning Procedures:**

Initialize $\phi, \psi, \theta$, target $\theta^{tgt} = \theta$, and replay buffer $\mathcal{R}$

**for** $n = 1, \cdots, N$ **do**

    Observe the initial state $\tilde{x}_0$

    **for** $k = 1, \cdots, K$ **do**

        Perform $a_{kh} \sim \mathcal{N}(\mu_\phi(\tilde{x}_{kh}), \sigma^2_{\text{explore}})$ and collect $r_{kh}, \tilde{x}_{(k+1)h}$

        Store $(\tilde{x}_{kh}, a_{kh}, r_{kh}, \tilde{x}_{(k+1)h})$ in $\mathcal{R}$

        **if** $k \equiv 0 \bmod m$ **then**

            ▷ *train advantage rate function and value function*

            Sample a batch of trajectories $\{\tilde{x}^{(i)}_{k_ih:(k_i+L)h}, a^{(i)}_{k_ih:(k_i+L)h}, r^{(i)}_{k_ih:(k_i+L)h}\}^B_{i=1}$ from $\mathcal{R}$

            Define $q_\psi(\tilde{x}, a) := \bar{q}_\psi(\tilde{x}, a) - \bar{q}_\psi(\tilde{x}, \mu_\phi(\tilde{x}))$

            Compute the martingale loss

$$\mathcal{L}^M = \frac{1}{B}\sum_{i=1}^B \left(V_\theta(\tilde{x}^{(i)}_{k_ih}) - \sum_{l=0}^{L-1} e^{-\beta lh}[r^{(i)}_{(k_i+l)h} - q_\psi(\tilde{x}^{(i)}_{(k_i+l)h}, a^{(i)}_{(k_i+l)h})]h - e^{-\beta Lh}V_{\theta^{tgt}}(\tilde{x}^{(i)}_{(k_i+L)h})\right)^2$$

$$(4.2)$$

            Sample a batch of terminal states $\{\tilde{x}^{(i)}_{Kh}, r^{(i)}_{Kh}\}^B_{i=1}$ from $\mathcal{R}$

            Compute the terminal value constraint $\mathcal{L}^C = \frac{1}{B}\sum_{i=1}^B (V_\theta(\tilde{x}^{(i)}_{Kh}) - r^{(i)}_{Kh})^2$

            Update the critic: $\psi \leftarrow \psi - \partial_\psi(\mathcal{L}^M + \alpha\mathcal{L}^C), \theta \leftarrow \theta - \eta\partial_\theta(\mathcal{L}^M + \alpha\mathcal{L}^C)$

            ▷ *train policy*

            Sample a batch of states $\{\tilde{x}^{(i)}_{k_ih}\}^B_{i=1}$ from $\mathcal{R}$

            Compute the policy loss $\mathcal{L} = -\frac{1}{B}\sum_{i=1}^B \bar{q}_\psi(\tilde{x}^{(i)}_{k_ih}, \mu_\phi(\tilde{x}^{(i)}_{k_ih}))h$

            Update the actor and target: $\phi \leftarrow \phi - \eta\partial_\phi\mathcal{L}$

            Update the target: $\theta^{tgt} \leftarrow \tau\theta + (1-\tau)\theta^{tgt}$

        **end if**

    **end for**

**end for**

---

**Remark 2** (effect of $1/h$ scaling). *Note that in Eq. (4.7), we omit the $1/h$ factor in contrast to Eq. (4.3). This modification is crucial for preventing the variance from blowing up. If we were to remove the $1/h$ factor in Eq. (4.3), then according to Prop. 4.1 the expected gradient $G_{\theta,h}$ would vanish as $h \to 0$. This theoretical inconsistency reveals a fundamental drawback of one-step TD methods in the continuous-time RL framework, which is also verified in our experiments.*

**Remark 3** (previous analysis of one-step TD). *Jia & Zhou (2022a) discussed one-step TD objective*

$$\min_\theta \frac{1}{h^2}\mathbb{E}_{\tilde{x}}\left(V_\theta(\tilde{x}_t) - r_t \cdot h - e^{-\beta h}V_\theta(\tilde{x}_{t+h})\right)^2, \qquad (4.10)$$

*showing that its minimizer does not converge to the true value function as $h \to 0$. However, practical one-step TD methods do not directly optimize Eq. (4.10), but rather employ the semi-gradient update Eq. (4.3). Consequently, the analysis in Jia & Zhou (2022a) does not fully explain the failure of discrete-time algorithms under small discretization steps. In contrast, our analysis is consistent with the actual update rule and thus offers theoretical insights of continuous-time algorithmic designs.*

## 5 EXPERIMENTS

The goal of our numerical experiments is to evaluate the efficiency of the proposed CT-DDPG algorithm and continuous-time RL framework in terms of convergence speed, training stability and robustness to the discretization step and dynamic noises.

**Environments.** We evaluate on a suite of challenging continuous-control benchmarks from Gymnasium (Towers et al., 2024): *Pendulum-v1*, *HalfCheetah-v5*, *Hopper-v5*, and *Walker2d-v5*, sweeping the discretization step and dynamic noise levels. To model stochastic dynamics, at each simulator step we sample an i.i.d. Gaussian generalized force $\xi \sim \mathcal{N}(0, \sigma^2 I)$ and write it to MuJoCo's `qfrc_applied` buffer (Todorov et al., 2012), thereby perturbing the equations of motion. More details can be found in Appendix D.

**Baselines.** We compare against discrete-time algorithms DDPG (Lillicrap et al., 2015), SAC (Haarnoja et al., 2018b), DDPG equipped with multi-step TD (Hessel et al., 2018) (see Appendix C for details), as well as a continuous-time algorithm with stochastic Gaussian policy: q-learning (Jia & Zhou, 2023). In particular, for q-learning, we adopt two different settings when learning q-function: the original one-step TD taregt ($L = 1$) in Jia & Zhou (2023), and a multi-step TD extension with $L > 1$ as in Alg. 1. This provides a fair comparison between deterministic and stochastic policies in continuous-time RL. We also test DAU (Tallec et al., 2019), i.e., CT-DDPG with $L = 1$, to see the effects of multi-step TD. For each algorithm, we report results averaged over at least three independent runs with different random seeds.

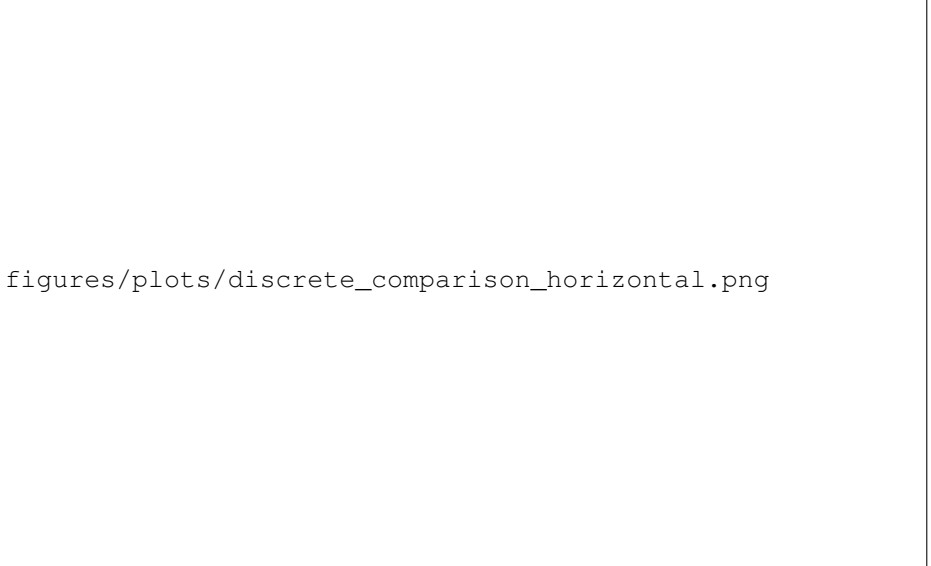

Figure 1: Comparison between CT-DDPG with discrete-time RL algorithms.

**Results.** Figs. 1 and 2 show the average return against training episodes, where the shaded area stands for standard deviation across different runs. We observe that for most environments, our CT-DDPG has the best performance among all baselines and the gap becomes larger as discretization step decreases and (or) noise level increases. Specifically, we have the following observations:

- As demonstrated in Fig. 1, although discrete-time algorithms, DDPG and SAC, perform reasonably well under the standard Gymnasium settings (top row), they degrade substantially when $h$ decreases and $\sigma$ increases (middle & bottom rows). This stems from the fact that one-step TD updates provide only myopic information under small $h$ and noisy dynamics, preventing the Q-function from capturing the long-term structure. Incorporating DDPG with multi-step TD can partially alleviate this issue; however, the method still exhibits slow convergence due to the persistent bias in the TD objective induced by off-policy sampling (see Appendix C for further discussions).

- For continuous-time RL with stochastic policy shown in Fig. 2, q-learning exhibits slow convergence and training instability, due to the difficulty of enforcing Bellman equation constraints Eq. (3.9). Although q-learning using multi-step TD can to some extent improve upon original q-learning ($L = 1$), it still remains unstable across diverse environment settings and underperforms compared to CT-DDPG. This highlights the fundamental limitations of stochastic policy in continuous-time RL.

- To further investigate the effects of multi-step TD, we also test DAU (i.e., CT-DDPG with $L = 1$) in Fig. 2. It turns out that in small $h$ and large $\sigma$ regime, DAU converges more slowly. In Fig. 3, we examine the variance to square norm ratio (NSR) of stochastic gradients in the training process. As $h \to 0$, NSR of DAU becomes evidently larger than that of CT-DDPG, consistent with our theories in Sec. 4.2. A large NSR leads to the instability when training q-function and consequently impedes the convergence.

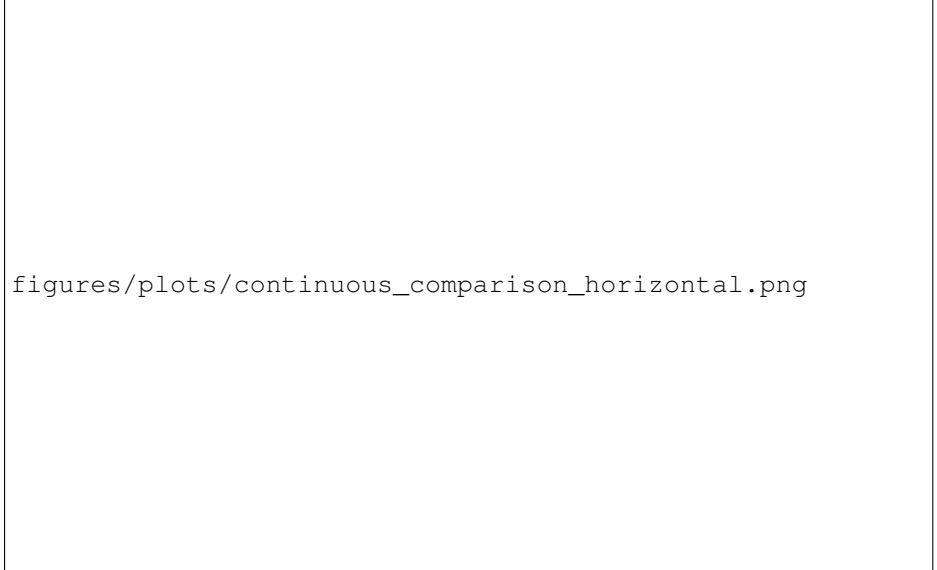

figures/plots/continuous_comparison_horizontal.png

Figure 2: Comparison between continuous-time RL algorithms.

figures/plots/nsr.png

Figure 3: Noise-to-Signal Ratio of stochastic gradient when training value-net.

**Ablations.** We further evaluate different choices of $L$ in CT-DDPG under noisy dynamics. The performance of choosing $L \propto 1/h$ remains robust across a wide range of $h$ and consistently outperforms the DAU baseline (i.e., $L = 1$), validating our theoretical results in Prop. 4.2. However, all choices of $L$ deteriorate when $h$ becomes extremely small, as the resulting state-transition noise dominates the dynamics, making the learning task substantially more challenging.

In summary, CT-DDPG exhibits superior performance in terms of convergence speed and stability across most environment settings, verifying the efficiency and robustness of our method.

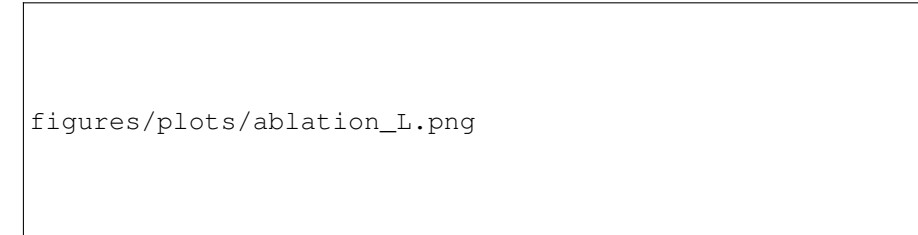

Figure 4: Ablations on $L$ in CT-DDPG.

## 6 CONCLUSION

In this paper, we investigate deterministic policy gradient methods to achieve stability and efficiency for deep RL in continuous-time environments, bridging the gap between discrete and continuous time algorithms. We develop a rigorous mathematical framework and provide a martingale characterization for DPG. We further theoretically demonstrate the issues of standard one-step TD method in continuous-time regime for the first time. All our theoretical results are verified through extensive experiments. We hope this work can motivate future researches on continuous-time RL.

**Ethics Statements.** This research does not involve human subjects, personally identifiable information, or sensitive data. All experiments are conducted on publicly available benchmark datasets under their respective licenses. We believe our work contributes positively to the research community by advancing fundamental understanding, and we encourage responsible and ethical applications of the proposed methods.

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

## A  RELATED WORK

**Discretization-Invariant Algorithms.**  Discretization has long been recognized as a central challenge in continuous control and RL (Baird, 1994; Doya, 2000; Munos, 2006). More recently, Tallec et al. (2019) showed that Q-learning–based approaches collapse as the discretization step becomes small and introduced the concept of the advantage rate function. Yildiz et al. (2021) tackled this issue through a model-based approach for deterministic ODE dynamics using the Neural ODE framework. Park et al. (2021) demonstrated that conventional policy gradient methods suffer from variance blowup and proposed action-repetition strategies as a remedy. Treven et al. (2023) systematically analyzed multiple discretization strategies. While these methods mitigate discretization sensitivity to some extent, they are restricted to deterministic dynamics and fail to handle stochasticity, a key feature of real-world environments. Treven et al. (2024) proposed an approach that jointly optimizes action and duration given interaction cost. Zhao et al. (2025a;b) established some theoretical analysis including sample efficiency and computational efficiency of model-based continuous-time RL methods.

**Continuous-Time RL with Stochastic Policies.**  Beyond addressing discretization sensitivity, another line of work directly considers continuous dynamics driven by stochastic differential equations. Jia & Zhou (2022a;b) introduced a martingale characterization for policy evaluation and developed an actor–critic algorithm in continuous time. Jia & Zhou (2023) studied the continuous-time analogue of the discrete-time advantage function, namely the $q$-function, and proposed a $q$-learning algorithm. Giegrich et al. (2024); Sethi et al. (2025) extend natural policy gradient methods to the continuous-time setting, and Zhao et al. (2023) further generalize PPO (Schulman et al., 2017) and TRPO (Schulman et al., 2015) methods to continuous time. However, all of these approaches adopt stochastic policies, which require enforcing Bellman equation constraints that are not tractable in deep RL frameworks. In contrast, our method leverages deterministic policies and enforces the Bellman equation via a simple reparameterization trick, enabling stable integration with deep RL.

**Theoretical Issues of Discrete-Time RL.**  Although many works have empirically observed that standard discrete-time algorithms degrade under small discretization, the theoretical foundations remain underexplored. Munos (2006); Park et al. (2021) showed that the variance of policy gradient estimators can diverge as $h \to 0$. Baird (1994); Tallec et al. (2019) further demonstrated that the standard Q-function degenerates and collapses to the value function. From the perspective of policy evaluation, Jia & Zhou (2022a) proved that the minimizer of the mean-square TD error does not converge to the true value function. Nevertheless, most discrete-time algorithms rely on semi-gradient updates rather than directly minimizing the mean-square TD error. To the best of our knowledge, there has been no theoretical analysis establishing the failure of standard one-step TD methods in the continuous-time setting.

## B  PROOFS

**Notations.**  We denote by $C^{1,2}([0,T] \times \mathbb{R}^n)$ the space of continuous functions $u : [0,T] \times \mathbb{R}^n \to \mathbb{R}$ that are once continuously differentiable in time and twice continuously differentiable in space, and there exists a constant $C \geq 0$ such that for all $(t,x) \in [0,T] \times \mathbb{R}^n$, $|u(t,x)| + |\partial_t u(t,x)| \leq C(1 + |x|^2), |\partial_x u(t,x)| \leq C(1 + |x|), |\partial_{xx}^2 u(t,x)| \leq C$. We use $\mathcal{P}(S)$ to denote the collection of all probability distributions over $S$. For compactness of notation, we denote by $\tilde{x}_t$ the concatenation of time and state $(t, x_t)$. Finally, we use standard $\mathcal{O}(\cdot), \Omega(\cdot), \Theta(\cdot)$ to omit constant factors.

### B.1  PROOF OF THM. 3.1

The following performance difference lemma characterizes the difference of value functions with different policies, which will be used in proving the policy gradient formula.

**Proposition B.1.** *Suppose Asp. 1 holds. Let $\phi \in \mathbb{R}^k$ and assume $V^\phi \in C^{1,2}([0,T] \times \mathbb{R}^n)$. For all $(t,x) \in [0,T] \times \mathbb{R}^n$ and $\phi' \in \mathbb{R}^k$,*

$$V^{\phi'}(t,x) - V^\phi(t,x)$$
$$= \mathbb{E}\left[ \int_t^T e^{-\beta(s-t)} \left( H[V^\phi](s, X_s^{\phi'}, \mu_{\phi'}(s, X_s^{\phi'})) - H[V^\phi](s, X_s^{\phi'}, \mu_\phi(s, X_s^{\phi'})) \right) ds \,\Big|\, X_t^{\phi'} = x \right]. \tag{B.1}$$

*Proof of Prop. B.1.* Observe that under Asp. 1, for each $\phi \in \mathbb{R}^k$, and $(t,x) \in [0,T] \times \mathbb{R}^n$,

$$V^\phi(t,x) := \mathbb{E}\left[ \int_t^T e^{-\beta(s-t)} r(s, X_s^{t,x,\phi}, \mu_\phi(s, X_s^{t,x,\phi})) dt + e^{-\beta(T-s)} g(X_T^{t,x,\phi}) \right], \tag{B.2}$$

where $(X_s^{t,x,\phi})_{s \geq t}$ satisfies for all $s \in [t,T]$,

$$dX_s = b(s, X_s, \mu_\phi(s, X_s)) ds + \sigma(s, X_s, \mu_\phi(s, X_s)) dW_s, \quad X_t = x. \tag{B.3}$$

Fix $\phi' \in \mathbb{R}^d$. Denote by $X^\phi = X^{t,x,\phi}$ and $X^{\phi'} = X^{t,x,\phi'}$ for simplicity. Then

$$V^{\phi'}(t,x) - V^\phi(t,x)$$
$$= \mathbb{E}\left[ \int_t^T e^{-\beta(s-t)} r(s, X_s^{\phi'}, \mu_{\phi'}(s, X_s^{\phi'})) ds \right] + e^{-\beta(T-t)} \mathbb{E}\left[ g(X_T^{\phi'}) \right] - V^\phi(t,x)$$
$$= \mathbb{E}\left[ \int_t^T e^{-\beta(s-t)} r(s, X_s^{\phi'}, \mu_{\phi'}(s, X_s^{\phi'})) ds \right] + \mathbb{E}\left[ e^{-\beta(T-t)} V^\phi(T, X_T^{\phi'}) \right] - V^\phi(t, X_t^{x,\phi'}), \tag{B.4}$$

where the last identity used the fact that $V^\phi(T,x) = g(x)$ and $X_t^{\phi'} = x$. As $V^\phi \in C^{1,2}([0,T] \times \mathbb{R}^n)$, applying Itô's formula to $s \mapsto e^{-\beta(s-t)} V^\phi(s, X_s^{\phi'})$ yields

$$\mathbb{E}\left[ e^{-\beta(T-t)} V^\phi(T, X_T^{\phi'}) \right] - V^\phi(t, X_t^{\phi'})$$
$$= \mathbb{E}\left[ \int_t^T e^{-\beta(s-t)} \mathcal{L}[V^\phi](s, X_s^{\phi'}, \mu_{\phi'}(s, X_s^{\phi'})) ds \right]$$
$$= \mathbb{E}\left[ \int_t^T e^{-\beta(s-t)} \left( \left( \mathcal{L}[V^\phi](s, y, \mu_{\phi'}(s, y)) - \mathcal{L}[V^\phi](s, y, \mu_\phi(s, y)) \right) \Big|_{y = X_s^{\phi'}} + \mathcal{L}[V^\phi](s, X_s^{\phi'}, \mu_\phi(s, X_s^{\phi'})) \right) ds \right]$$
$$= \mathbb{E}\left[ \int_t^T e^{-\beta(s-t)} \left( \left( \mathcal{L}[V^\phi](s, y, \mu_{\phi'}(s, y)) - \mathcal{L}[V^\phi](s, y, \mu_\phi(s, y)) \right) \Big|_{y = X_s^{\phi'}} - r(s, X_s^{\phi'}, \mu_\phi(s, X_s^{\phi'})) \right) ds \right],$$

where the last identity used the PDE Eq. (3.2). This along with Eq. (B.4) proves the desired result. $\square$

*Proof of Thm. 3.1.* Recall that $\partial_\phi V^\phi(t,x) = (\partial_{\phi_1} V^\phi(t,x), \ldots, \partial_{\phi_k} V^\phi(t,x))^\top$. Hence it suffices to prove for all $\phi' \in \mathbb{R}^k$,

$$\frac{d}{d\epsilon} V^{\phi + \epsilon\phi'}(t,x) \Big|_{\epsilon=0} = \mathbb{E}\left[ \int_t^T e^{-\beta(s-t)} \partial_a A^\phi(s, X_s^\phi, \mu_\phi(s, X_s^\phi))^\top \partial_\phi \mu_\phi(s, X_s^\phi) ds \,\Big|\, X_t^\phi = x \right] \phi'.$$

To this end, for all $\epsilon \in [-1,1]$, let $X^\epsilon$ be the solution to the following dynamics:

$$dX_s = b(s, X_s, \mu_{\phi+\epsilon\phi'}(s, X_s)) ds + \sigma(s, X_s, \mu_{\phi+\epsilon\phi'}(t, X_t)) dW_s, \quad X_t = x. \tag{B.5}$$

For all $\epsilon \in [-1,1]$, by Prop. B.1 and the fundamental theorem of calculus,

$$\frac{V^{\phi+\epsilon\phi'}(t,x) - V^\phi(t,x)}{\epsilon} = \mathbb{E}\left[ \int_t^T e^{-\beta(s-t)} \left( \int_0^1 \mathcal{G}(s, X_s^\epsilon, \phi + r\epsilon\phi') dr \right) ds \right] \phi', \tag{B.6}$$

where for all $\tilde{\phi} \in \mathbb{R}^k$,

$$\mathcal{G}(t, x, \tilde{\phi}) := \partial_a H[V^\phi](t, x, \mu_{\tilde{\phi}}(s, x))^\top \partial_\phi \mu_{\tilde{\phi}}(t, x).$$

To show the limit of Eq. (B.6) as $\epsilon \to 0$, observe that by Asp. 2 and standard stability analysis of Eq. (B.5) (see e.g., (Zhang, 2017, Theorem 3.2.4)), for all $\epsilon \in [-1, 1]$,

$$\mathbb{E}\left[\sup_{t \le s \le T} |X_s^\epsilon - X_s^0|^2\right] \le C\mathbb{E}\left[\left(\int_0^T |b(s, X_s^0, \mu_{\phi+\epsilon\phi'}(s, X_s^0)) - b(s, X_s^0, \mu_\phi(s, X_s^0))|\mathrm{d}s\right)^2\right]$$
$$+ C\mathbb{E}\left[\int_0^T |\sigma(s, X_s^0, \mu_{\phi+\epsilon\phi'}(s, X_s^0)) - \sigma(s, X_s^0, \mu_\phi(s, X_s^0))|^2\mathrm{d}s\right],$$

which along with the growth condition in Asp. 1 and the regularity of $b$, $\sigma$ and $\phi$ in Asp. 2, and the dominated convergence theorem shows that

$$\lim_{\epsilon \to 0} \mathbb{E}\left[\sup_{t \le s \le T} |X_s^\epsilon - X_s^0|^2\right] = 0. \tag{B.7}$$

Moreover, there exists $C \ge 0$ such that for all $\epsilon \in [-1, 1]$, and $A \in \mathcal{F} \otimes \mathcal{B}([0, T]) \otimes \mathcal{B}([0, 1])$,

$$\mathbb{E}\left[\int_t^T \int_0^1 \mathbf{1}_A e^{-\beta(s-t)} |\mathcal{G}(s, X_s^\epsilon, \phi + r\epsilon\phi')| \,\mathrm{d}r\mathrm{d}s\right]$$
$$\le \mathbb{E}\left[\int_t^T \int_0^1 \mathbf{1}_A \mathrm{d}r\mathrm{d}s\right]^{\frac{1}{2}} \mathbb{E}\left[\int_t^T \int_0^1 e^{-2\beta(s-t)} |\mathcal{G}(s, X_s^\epsilon, \phi + r\epsilon\phi')|^2 \,\mathrm{d}r\mathrm{d}s\right]^{\frac{1}{2}}$$
$$\le \mathbb{E}\left[\int_t^T \int_0^1 \mathbf{1}_A \mathrm{d}r\mathrm{d}s\right]^{\frac{1}{2}} C \left(1 + \mathbb{E}\left[\sup_{t \le s \le T} |X_s^\epsilon|^2\right]\right)^{\frac{1}{2}},$$

where the last inequality used the growth conditions on the derivatives of the coefficients $b, \sigma, r$ and $\mu$, and of the value function $V^\phi$. Using the moment condition $\sup_{\epsilon \in [-1,1]} \mathbb{E}\left[\sup_{t \le s \le T} |X_s^\epsilon|^2\right] < \infty$, the random variables $\{(\omega, s, r) \mapsto e^{-\beta(s-t)}\mathcal{G}(s, X_s^\epsilon, \phi + r\epsilon\phi') \mid \epsilon \in [-1, 1]\}$ are uniformly integrable. Hence using Vitali's convergence theorem and passing $\epsilon \to 0$ in Eq. (B.6) yield the desired identity. $\qquad\square$

## B.2 PROOF OF THM. 3.2

*Proof.* For all $(t, x) \in [0, T] \times \mathbb{R}^n$ and $a \in \mathcal{O}_{\mu_\phi(t,x)}$, applying Itô's formula to $u \mapsto e^{-\beta(u-t)}\hat{V}(s, X_u^{t,x,a})$ yields for all $0 \le t < s \le T$,

$$e^{-\beta(s-t)}\hat{V}(s, X_s^{t,x,a}) - \hat{V}(t, X_t^{t,x,a}) = \int_t^s e^{-\beta(u-t)}\mathcal{L}[\hat{V}](u, X_u^{t,x,a}, \alpha_u)\mathrm{d}u$$
$$+ \int_t^s e^{-\beta(u-t)}\partial_x\hat{V}(u, X_u^{t,x,a})^\top \sigma(u, X_u^{t,x,a}, \alpha_u)\mathrm{d}W_u. \tag{B.8}$$

This along with the martingale condition Eq. (3.7) implies

$$\left(\int_t^s e^{-\beta(u-t)}\left(\mathcal{L}[\hat{V}] + r - \hat{q}\right)(u, X_u^{t,x,a}, \alpha_u)\mathrm{d}u\right)_{s \in [t,T]}$$

is a martingale, which has continuous paths and finite variation. Hence almost surely

$$\int_t^s e^{-\beta(u-t)}\left(\mathcal{L}[\hat{V}] + r - \hat{q}\right)(u, X_u^{t,x,a}, \alpha_u)\mathrm{d}u = 0, \quad \forall s \in [t, T]. \tag{B.9}$$

We claim $(\mathcal{L}[\hat{V}] + r - \hat{q})(t, x, a) = 0$ for all $(t, x) \in [0, T] \times \mathbb{R}^n$ and $a \in \mathcal{O}_{\mu_\phi(t,x)}$. To see it, define $f(t, x, a) := (L[\hat{V}] + r - \hat{q})(t, x, a)$ for all $(t, x, a) \in [0, T] \times \mathbb{R}^n \times \mathcal{A}$. By assumptions,

$f \in C([0, T] \times \mathbb{R}^n \times \mathcal{A})$. Suppose there exists $(\bar{t}, \bar{x}) \in [0, T] \times \mathbb{R}^n$ and $\bar{a} \in \mathcal{O}_{\mu_\phi(t,x)}$ such that $f(\bar{t}, \bar{x}, \bar{a}) \neq 0$. Due to the continuity of $f$, we can assume without loss of generality that $f(\bar{t}, \bar{x}, \bar{a}) > 0$ and $\bar{t} \in [0, T)$. The continuity of $f$ implies that there exist constants $\epsilon, \delta > 0$ such that $f(t, x, a) \geq \epsilon > 0$ for all $(t, x, a) \in [0, T] \times \mathbb{R}^n \times \mathcal{A}$ with $\max\{|t - \bar{t}|, |x - \bar{x}|, |a - \bar{a}|\} \leq \delta$. Now consider the process $X^{\bar{t}, \bar{x}, \bar{a}}$ defined by (3.8), and define the stopping time

$$\tau := \inf \left\{ t \in [\bar{t}, T] \mid \max\{|t - \bar{t}|, |X_t^{\bar{t}, \bar{x}, \bar{a}} - \bar{x}|, |\alpha_t - \bar{a}|\} > \delta \right\}.$$

Note that $\tau > \bar{t}$ almost surely, due to the sample path continuity of $t \mapsto X_t^{\bar{t}, \bar{x}, \bar{a}}$ and the condition $\lim_{s \searrow t} \alpha_s = \bar{a}$. This along with (B.9) implies that there exists a measure zero set $\mathcal{N}$ such that for all $\omega \in \Omega \setminus \mathcal{N}, \tau(\omega) > \bar{t}$, and

$$\int_{\bar{t}}^{\tau(\omega)} e^{-\beta(u-\bar{t})} f\left(u, X_u^{\bar{t}, \bar{x}, \bar{a}}(\omega), \alpha_u(\omega)\right) \mathrm{d}u = 0.$$

However, by the definition of $\tau$, for all $t \in (\bar{t}, \tau(\omega))$, $\max\{|t - \bar{t}|, |X_t^{\bar{t}, \bar{x}, \bar{a}} - \bar{x}|, |\alpha_t - \bar{a}|\} \leq \delta$, which along with the choice of $\delta$ implies $f(t, X_t^{\bar{t}, \bar{x}, \bar{a}}(\omega), \alpha_t(\omega)) \geq \epsilon > 0$ and hence

$$\int_{\bar{t}}^{\tau(\omega)} e^{-\beta(u-\bar{t})} h\left(u, X_u^{\bar{t}, \bar{x}, \bar{a}}(\omega), \alpha_u(\omega)\right) \mathrm{d}u > 0.$$

This yields a contradiction, and proves $(\mathcal{L}[\hat{V}] + r - \hat{q})(t, x, a) = 0$ for all $(t, x, \mu) \in [0, T] \times \mathbb{R}^n$ and $a \in \mathcal{O}_{\mu_\phi(t,x)}$.

Now by Eq. (3.6), for all $(t, x) \in [0, T] \times \mathbb{R}^n$,

$$(\mathcal{L}[\hat{V}] + r)(t, x, \mu_\phi(t, x)) = 0, \quad \hat{V}(T, x) = g(x).$$

Since $V^\phi \in C^{1,2}([0, T] \times \mathbb{R}^n)$ satisfies the same PDE, the uniqueness of Feynman-Kac formula (Beck et al., 2021) shows that $\hat{V}(t, x) = V^\phi(t, x)$ for all $(t, x)$. This subsequently implies $(\mathcal{L}[V^\phi] + r - \hat{q})(t, x, a) = 0$ for all $(t, x) \in [0, T] \times \mathbb{R}^n$ and $a \in \mathcal{O}_{\mu_\phi(t,x)}$. □

### B.3 PROOF OF PROP. 4.1

*Proof.* By Itô's formula,

$$e^{-\beta h} V_\theta(\tilde{x}_{t+h}) - V_\theta(\tilde{x}_t)$$
$$= \underbrace{\int_t^{t+h} e^{-\beta(s-t)} \left[ \partial_t V_\theta(\tilde{x}_s) + \partial_x V_\theta(\tilde{x}_s)^\top b(\tilde{x}_s, a_s) + \frac{1}{2} \mathrm{Tr}(\partial_{xx}^2 V_\theta(\tilde{x}_s) \sigma \sigma^\top(\tilde{x}_s, a_s)) - \beta V_\theta(\tilde{x}_s) \right] \mathrm{d}s}_{①}$$
$$+ \underbrace{\int_t^{t+h} e^{-\beta(s-t)} \partial_x V_\theta(\tilde{x}_s)^\top \sigma(\tilde{x}_s, a_s) \mathrm{d}W_s}_{②}.$$

(B.10)

Note that the last term is a martingale and thus vanishes after taking expectation. Therefore the semi-gradient can be rewritten as

$$G_{\theta,h} = \mathbb{E}\left[ \partial_\theta V_\theta(\tilde{x}_t) \left( -① \cdot \frac{1}{h} + (A_\psi(\tilde{x}_t, a_t) - r_t) \right) \right].$$

(B.11)

When the discretization step $h$ goes to zero, the integral ① admits a first-order expansion, which leads to

$$\lim_{h \to 0} G_{\theta,h} = \mathbb{E}\left[ \partial_\theta V_\theta(\tilde{x}_t) \left( A_\psi(\tilde{x}_t, a_t) - \partial_t V_\theta(\tilde{x}_t) - H(\tilde{x}_t, a_t, \partial_x V_\theta(\tilde{x}_t), \partial_{xx}^2 V_\theta(\tilde{x}_t)) + \beta V_\theta(\tilde{x}_t) \right) \right].$$

(B.12)

Similarly we have

$$\lim_{h \to 0} G_{\psi,h} = \mathbb{E}\left[ \partial_\psi A_\psi(\tilde{x}_t, a_t) \left( A_\psi(\tilde{x}_t, a_t) - \partial_t V_\theta(\tilde{x}_t) - H(\tilde{x}_t, a_t, \partial_x V_\theta(\tilde{x}_t), \partial_{xx}^2 V_\theta(\tilde{x}_t)) + \beta V_\theta(\tilde{x}_t) \right) \right].$$

(B.13)

On the other hand, consider the conditional variance of stochastic gradient:

$$\mathrm{Var}(g_{\theta,h} \mid \mathcal{F}_t) = \frac{1}{h^2} \partial_\theta V_\theta(\tilde{x}_t) \partial_\theta V_\theta(\tilde{x}_t)^\top \mathrm{Var}(e^{-\beta h} V_\theta(\tilde{x}_{t+h}) - V_\theta(\tilde{x}_t) \mid \mathcal{F}_t). \tag{B.14}$$

Note that

$$\mathbb{E}[(e^{-\beta h} V_\theta(\tilde{x}_{t+h}) - V_\theta(\tilde{x}_t))^2 \mid \mathcal{F}_t] = \mathbb{E}[①^2 + 2 \cdot ① \cdot ② + ②^2 \mid \mathcal{F}_t], \tag{B.15}$$

and $\mathbb{E}[e^{-\beta h} V_\theta(\tilde{x}_{t+h}) - V_\theta(\tilde{x}_t) \mid \mathcal{F}_t] = \mathbb{E}[① \mid \mathcal{F}_t]$. This yields

$$\mathrm{Var}(e^{-\beta h} V_\theta(\tilde{x}_{t+h}) - V_\theta(\tilde{x}_t) \mid \mathcal{F}_t) = \mathrm{Var}(① \mid \mathcal{F}_t) + \mathbb{E}\left[②^2 + 2 \cdot ① \cdot ② \mid \mathcal{F}_t\right] \geq \mathbb{E}\left[②^2 + 2 \cdot ① \cdot ② \mid \mathcal{F}_t\right] \tag{B.16}$$

According to Itô isometry,

$$\mathbb{E}[①^2 \mid \mathcal{F}_t] = \mathcal{O}(h^2), \tag{B.17}$$

$$\mathbb{E}[②^2 \mid \mathcal{F}_t] = \mathbb{E}\left[\int_t^{t+h} e^{-2\beta(s-t)} \|\partial_x V_\theta(\tilde{x}_s)^\top \sigma(\tilde{x}_s, a_s)\|^2 \mathrm{d}s \,\middle|\, \tilde{x}_t\right] = \mathcal{O}(h), \tag{B.18}$$

and the cross term can be controlled by Cauchy-Schwarz:

$$\mathbb{E}[|① \cdot ②| \mid \tilde{x}_t] \leq (\mathbb{E}[①^2 \mid \tilde{x}_t])^{\frac{1}{2}} \cdot (\mathbb{E}[②^2 \mid \tilde{x}_t])^{\frac{1}{2}} = \mathcal{O}(h^{\frac{3}{2}}) \tag{B.19}$$

These estimates show that, as $h \to 0$, the leading contribution to the variance comes from the stochastic integral term ②. As a result, by combining Fatou's Lemma and Eq. (B.14), we conclude that

$$\begin{aligned}
\lim_{h \to 0} h \cdot \mathrm{Var}(g_{\theta,h}) &\geq \lim_{h \to 0} h \cdot \mathbb{E}[\mathrm{Var}(g_{\theta,h} \mid \mathcal{F}_t)] \\
&\geq \mathbb{E}\left[\lim_{h \to 0}[h \cdot \mathrm{Var}(g_{\theta,h} \mid \mathcal{F}_t)]\right] \\
&\geq \mathbb{E}\left[\partial_\theta V_\theta(\tilde{x}_t) \partial_\theta V_\theta(\tilde{x}_t)^\top \|\partial_x V_\theta(\tilde{x}_t)^\top \sigma(\tilde{x}_t, a_t)\|^2\right].
\end{aligned} \tag{B.20}$$

$\square$

### B.4 PROOF OF PROP. 4.2

*Proof.* We begin by recalling that, for any horizon $Lh$, Itô's formula yields,

$$e^{-\beta Lh} V_\theta(\tilde{x}_{t+Lh}) - V_\theta(\tilde{x}_t)$$
$$= \underbrace{\int_t^{t+Lh} e^{-\beta(s-t)} \left[\partial_t V_\theta(\tilde{x}_s) + \partial_x V_\theta(\tilde{x}_s)^\top b(\tilde{x}_s, a_s) + \frac{1}{2}\mathrm{Tr}(\partial_{xx}^2 V_\theta(\tilde{x}_s) \sigma\sigma^\top(\tilde{x}_s, a_s)) - \beta V_\theta(\tilde{x}_s)\right] \mathrm{d}s}_{③}$$
$$+ \underbrace{\int_t^{t+Lh} e^{-\beta(s-t)} \partial_x V_\theta(\tilde{x}_s)^\top \sigma(\tilde{x}_s, a_s) \mathrm{d}W_s}_{④}. \tag{B.21}$$

Now consider the case where $Lh \equiv \delta > 0$ is fixed while $h \to 0$. In this regime, the estimator $G_{\theta,h,\delta/h}$ can be expressed as

$$\begin{aligned}
\lim_{h \to 0} G_{\theta,h,\frac{\delta}{h}} &= \mathbb{E}\left[\partial_\theta V_\theta(\tilde{x}_t)\left(V_\theta(\tilde{x}_t) - \int_t^{t+\delta} e^{-\beta(s-t)}[r_s - q_\psi(\tilde{x}_s, a_s)]\mathrm{d}s - e^{-\beta\delta} V_\theta(\tilde{x}_{t+\delta})\right)\right] = \Theta(1) \\
&= \mathbb{E}\left[\partial_\theta V_\theta(\tilde{x}_t)\left(\int_t^{t+\delta}\left[q_\psi(\tilde{x}_s, a_s) - \partial_t V_\theta(\tilde{x}_s) - H(\tilde{x}_s, a_s, \partial_x V_\theta(\tilde{x}_s), \partial_{xx}^2 V_\theta(\tilde{x}_s)) + \beta V_\theta(\tilde{x}_s)\right]\mathrm{d}s\right)\right] \\
&= \Theta(1).
\end{aligned} \tag{B.22}$$

The integral is taken over a fixed interval of length $\delta$, and thus this expression is bounded and will not vanish.

We next turn to the variance. Expanding the definition of $g_{\theta,h,L}$ and using Jensen's inequality, we obtain

$$\text{Var}(g_{\theta,h,L})$$

$$\leq 2\mathbb{E}\left[\partial_\theta V_\theta(\tilde{x}_t)\partial_\theta V_\theta(\tilde{x}_t)^\top \left((e^{-\beta Lh}V_\theta(\tilde{x}_{t+Lh}) - V_\theta(\tilde{x}_t))^2 + (\sum_{l=0}^{L-1} e^{-\beta lh}[r_{t+lh} - q_\psi(\tilde{x}_{t+lh}, a_{t+lh})]h)^2\right)\right]$$

$$= \mathcal{O}(1).$$
(B.23)

This is because all terms are bounded. □

## C  DISCUSSIONS ON MULTI-STEP TD IN OFF-POLICY DISCRETE-TIME RL

Discrete algorithms can also be combined with multi-step TD as in Hessel et al. (2018). Given a stochastic policy $\pi(\cdot|\cdot)$ (possibly deterministic) and a horizon $L > 1$, the TD($L$) objective is defined as

$$\min_\psi \mathbb{E}\left(Q_\psi(\tilde{x}_t, a_t) - \sum_{k=0}^{L-1} e^{-\beta kh}r(\tilde{x}_{t+kh}, a_{t+kh}) - e^{-\beta Lh}Q_\psi(\tilde{x}_{t+Lh}, a_{t+Lh})\right)^2.$$
(C.1)

By definition of the Q-function, Eq. (C.1) is valid only if $a_{t+kh} \sim \pi(\cdot|\tilde{x}_{t+kh})$ for any $k \geq 1$. However, in off-policy setting, the trajectory $\{(\tilde{x}_{t+kh}, \tilde{a}_{t+kh})\}_{1 \leq k \leq L-1}$ is drawn from the replay buffer generated under another exploration policy $\pi'(\cdot|\cdot)$ (or possibly multiple exploration policies). This mismatch induces bias when optimizing Eq. (C.1) with $L > 1$. Note that there is no such bias when $L = 1$ since $a_t$ doesn't need to be sampled from $\pi(\cdot|\tilde{x}_t)$ Therefore, naively combining multi-step TD with experience replay in off-policy discrete-time algorithms such as DDPG and SAC introduces an extra source of bias, in contrast to continuous-time algorithms.

## D  EXPERIMENT DETAILS

**Model architecture.**  Across all experiments, the policy, Q-network, and value network are implemented as three-layer fully connected MLPs with ReLU activations. The hidden dimension is set to 400, except for *Pendulum*, where we use 64. To incorporate time information, we augment the environment observations with a sinusoidal embedding, yielding $\tilde{x}_t = (x_t, \cos(\frac{2\pi t}{T}), \sin(\frac{2\pi t}{T}))$, where $T$ denotes the maximum horizon. For stochastic policies, we employ Gaussian policies with mean and variance parameterized by neural networks, and fix the entropy coefficient to $\gamma = 0.1$.

**Environment setup.**  To accelerate training, we run 8 environments in parallel, collecting 8 trajectories per episode. The discount rate is set to $\beta = 0.8$, applied in the form $e^{-\beta h}$. For MuJoCo environments, we set `terminate_when_unhealthy=False`.

**Training hyperparameters.**  We use the Adam optimizer with a learning rate of $3 \times 10^{-4}$ for all networks ($3 \times 10^{-3}$ for *Pendulum*), and a batch size of $B = 256$. The update frequency is $m = 1$ in the original environment and $m = 5$ for smaller step sizes $h$. The soft target update parameter is $\tau = 0.005$. The weight for the terminal value constraint is $\alpha = 0.002$. For CT-DDPG, the trajectory length $L$ is sampled uniformly from $[2, 10]$, and we use exploration noise with standard deviation $\sigma_{\text{explore}} = 0.1$. For q-learning, for each state $\tilde{x}$ in the minibatch, we sample $n = 20$ actions from $\pi(\cdot \mid \tilde{x})$ and compute the penalty term $\left(\frac{1}{n}\sum_{i=1}^n \left[q_\psi(\tilde{x}, a_i) - \gamma \log \pi(a_i \mid \tilde{x})\right]\right)^2$.

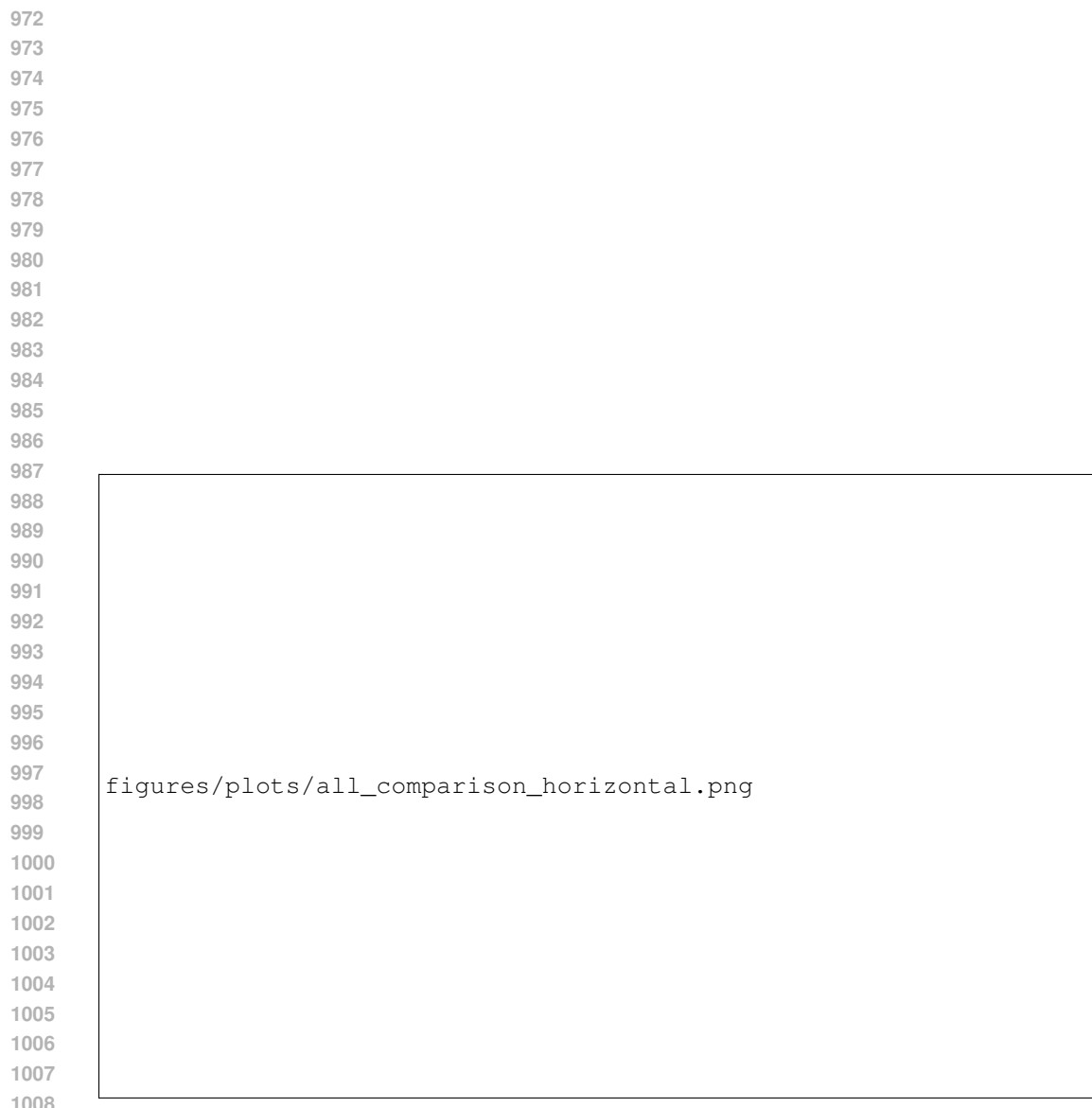

Figure 5: Comparison between all algorithms.

