# OpenReview forum: "Bridging Discrete and Continuous RL: Stable Deterministic Policy Gradient with Martingale Characterization"
_ICLR.cc/2026/Conference — Submitted to ICLR 2026_

### Official Review · Reviewer_C2As · 2025-10-31

**Soundness:** 2
**Presentation:** 2
**Contribution:** 2
**Rating:** 2
**Confidence:** 4

**Summary:**

This paper proposes a principled framework for deterministic policy gradient (DPG) in continuous-time reinforcement learning (CTRL). The authors derive a continuous-time analogue of the advantage-based policy gradient theorem and establish its martingale characterization, providing a theoretical foundation for stable deterministic policy optimization in continuous domains. Building upon this theory, they introduce a new algorithm, CT-DDPG, designed to mitigate the instability and discretization sensitivity often observed in discrete-time RL algorithms when applied to continuous dynamical systems. Experimental evaluations on several continuous control benchmarks demonstrate improved convergence stability and learning efficiency compared to both discrete-time and prior continuous-time baselines. Overall, the paper aims to bridge the theoretical and algorithmic gap between discrete-time RL methods and real-world continuous control systems.

**Strengths:**

1. The paper establishes a mathematically solid foundation for deterministic policy gradient (DPG) methods in continuous-time reinforcement learning. The authors derive a continuous-time analogue of the advantage function (Theorem 3.1) and rigorously prove its martingale characterization (Theorem 3.2), connecting the policy gradient to the advantage-rate function under deterministic policies. This framework generalizes the discrete-time DPG results (Silver et al., 2014) while removing restrictive assumptions such as uniform ellipticity and purely stochastic policies. The martingale-based formulation elegantly guarantees consistency of the value and advantage functions and provides a clear theoretical pathway for algorithm design.
2. The paper directly targets the issue that discrete-time RL algorithms become unstable as the time-step $h\to0$. The authors provide a precise theoretical explanation for this degradation: one-step temporal-difference (TD) updates cause gradient variance to blow up (Proposition 4.1). To overcome this, they introduce the Continuous-Time Deep Deterministic Policy Gradient (CT-DDPG) algorithm using multi-step TD objectives (Eq. 4.7), proving its variance remains bounded (Proposition 4.2).

**Weaknesses:**

1. The paper’s theoretical framing and algorithmic contribution appear incremental relative to prior work in continuous-time reinforcement learning (CTRL). While it emphasizes the use of a martingale characterization to derive a deterministic policy gradient (DPG) formula, the idea is conceptually close to existing stochastic-policy frameworks (Jia & Zhou, 2022a; 2022b; 2023) and the model-based DPG formulations of [5]. The paper does not convincingly clarify why the martingale representation provides a fundamentally new perspective or practical advantage over Itô-based stochastic analysis or existing actor–critic formulations.

   Moreover, the literature review is narrow and overlooks several recent studies towards same target. For example: [1] systematically analyzes multiple discretization strategies (MSS) for model-based CTRL; [2] proposes a time-adaptive sensing and control approach that jointly optimizes action and duration, directly tackling the same step-size sensitivity problem that this paper aims to address; [3] focuses on improving sample and computational efficiency while maintaining performance; and [4] theoretically studies how, for stochastic dynamics, the observation interval should adapt to the system’s variance. Although the authors cite [5], the discussion remains superficial—[5] already develops a continuous-time actor–critic framework that learns a deterministic policy gradient under ODE dynamics, conceptually similar to this work.

   The paper does not clearly articulate how its CT-DDPG algorithm differs from or improves upon these established baselines, many of which already achieve comparable goals such as variance control, discretization invariance, and stability. A more thorough comparative discussion and empirical ablation referencing these works would be necessary to substantiate the claimed contributions.

   [1] Treven, L., Hübotter, J., Sukhija, B., Dorfler, F., & Krause, A. (2023). Efficient exploration in continuous-time model-based reinforcement learning. _Advances in Neural Information Processing Systems_, _36_, 42119-42147.

   [2] Treven, Lenart, et al. "When to sense and control? a time-adaptive approach for continuous-time RL." _Advances in Neural Information Processing Systems_ 37 (2024): 63654-63685.

   [3] Zhao, R., Yu, Y., Zhu, A. Y., Yang, C., & Zhou, D. Sample and Computationally Efficient Continuous-Time Reinforcement Learning with General Function Approximation. In _The 41st Conference on Uncertainty in Artificial Intelligence_.

   [4] Zhao, R., Yu, Y., Wang, R., Huang, C., & Zhou, D. (2025). Instance-Dependent Continuous-Time Reinforcement Learning via Maximum Likelihood Estimation. _arXiv preprint arXiv:2508.02103_.

   [5] Yildiz, C., Heinonen, M., & Lähdesmäki, H. (2021, July). Continuous-time model-based reinforcement learning. In _International Conference on Machine Learning_ (pp. 12009-12018). PMLR.

2. The set of stochastic baselines is insufficient. It is unclear why the paper does not include comparisons with ctrl methods such as those proposed in [2] and [5]. In Figure 2, the only continuous-time baseline shown is the q-learning algorithm. Since the authors already cite [5] in the main paper, it would be natural—and important—to include it as a baseline to better demonstrate the claimed advantages of the proposed deterministic approach.
3. The reported results also appear somewhat counter-intuitive. In Figure 2(c) (Hopper, h = 0.008, $\sigma$ = 0), the q-learning (L = 1) baseline exhibits a substantially larger error bar than in panels (g) and (k), even though the noise level is lower. This behavior is unexpected and not explained in the paper—it raises questions about potential implementation differences, instability in training, or inconsistent experimental conditions.

**Questions:**

Please compare with the papers listed in the weakness part. I would be happy to raise the score if the authors can address the concerns as listed in the weakness part.

---

> ### Author Response · Authors · 2025-11-26
>
> Thanks for your careful review and constructive feedback! We address your concerns as follows.
>
> **W1**: The idea is conceptually close to existing stochastic-policy frameworks (Jia & Zhou, 2022a; 2022b; 2023) and the model-based DPG formulations of [5]. The paper does not convincingly clarify why the martingale representation provides a fundamentally new perspective or practical advantage over Itô-based stochastic analysis or existing actor–critic formulations.
>
> **A1**:
>
> (i) We would like to emphasize that our work, based on a deterministic policy, represents a significant conceptual departure from earlier studies on stochastic policies. While it is well recognized that the ultimate goal in RL is to learn a deterministic policy, previous works (e.g., by Jia and Zhou) primarily focused on stochastic policies, often as a consequence of using exploratory SDE settings with entropy regularization. Critically, the feasibility of the stochastic policy approach in continuous-time and continuous-state RL is largely restricted to the Linear-Quadratic Regulator (LQR) case, where the optimal policy is Gaussian, allowing one to focus solely on its mean. The major drawback of this approach for general RL problems is the difficulty in sampling a stochastic policy from the optimal Gibbs measure, making the subsequent retrieval of a deterministic policy nearly impossible. In contrast, our framework and analysis demonstrate, for the first time, how to design efficient algorithms to learn the deterministic policy directly in continuous-time RL problems.
>
>
> (ii) Technically,  Thm 3.2 is fundamentally different from the martingale results of Thm 7 in (Jia and Zhou, 2023), despite some similarity in their appearance. Thm 7 in (Jia and Zhou, 2023) can only characterize the advantage-rate function for ***fully supported*** stochastic policy and imposes the Bellman condition $\mathbb{E}_{a\sim \pi(a|t,x)}[\hat q(t,x,a)-\gamma\log\pi(a|t,x)] = 0$, which is impossible to satisfy for general neural networks. In contrast, our Thm 3.2 provides a martingale characerization for deterministic policy, which can be readily plugged in off-policy settings. In addition, the Bellman condition in Eq(3.6) can be easily satisfied via a reparameterization trick Eq(4.1), leading to the fundamental benefits of learning deterministic policy compared to (Jia and Zhou, 2023).
>
> (iii) Moreover, Thm 3.1 is novel since it removes most stringent conditions in existing works as discussed in line 163-165. In contrast, we don not see any DPG formula in [5].
>
> (iv) We thank the reviewer for mentioning references [1-5] on continuous-time RL. Nevertheless, in this paper we focus on **model-free** RL algorithms, while [1,3,4,5] are all **model-based** methods, which are **conceptually different and of different focus**. This is primarily because the model-based methods needs to first learn the transition dynamics and then requires an additional oracle to solve the control problem based on the estimated dynamic, fundamentally distinct from the principle of model-free approach. [2] studies how to select interaction frequency in continuous-time RL given interaction cost, while in this paper we assume the interaction frequency has been determined by the system as a prior. Overall, we emphasize that the goal of this paper is to **develop a theoretically grounded (model-free) algorithm that achieves stability and efficiency for deep RL in (nearly) continuous-time environments**. To make the disctinction, we have added detailed  discussion regarding  these references in Appendix A in the updated version.
>
> **W2**: The set of stochastic baselines is insufficient. It is unclear why the paper does not include comparisons with ctrl methods such as those proposed in [2] and [5].
>
> **A2**: As mentioned in A1, this paper focuses on model-free algorithms while [1,3,4,5] are all model-based methods and of different focus. Moreover,  [2] studies how to select interaction frequency given interaction cost while we assumt this is determinied by the system as a prior. Still, we evaluate an additional baseline, DDPG with multi-step TD, to examine the performance of discrete-time algorithms when combined with multi-step TD. The results are reported in Fig. 1 in the updated version and more discussions are provided in Appendix C. These new results show that naively combining multi-step TD with experience replay in off-policy discrete-time algorithms such as DDPG and SAC introduces an extra source of bias, and suffer from slow convergence compared to continuous-time approaches.

---

> ### Author Response · Authors · 2025-11-26
>
> **W3**: In Figure 2(c) (Hopper, $h = 0.008, \sigma = 0$), the q-learning (L = 1) baseline exhibits a substantially larger error bar than in panels (g) and (k), even though the noise level is lower. This behavior is unexpected and not explained in the paper.
>
> **A3**: The large error bar in Fig. 2(c,g,k) is not because of the noise level of dynamics, instead it is  **primarily due to the instability of q-learning, and more specifically, the impracticality of the Bellman condition Eq(3.9) (in contrast to Eq(3.6))**.
>
> Thank you for your remark. We would like to address this comments from several aspects.
> (i) First, let us emphasize that the Bellman condition is crucial in martingale characterization because it is the only condition to ensure that $\hat{V},\hat{q}$ holds for the current policy. Without this Bellman condition,  the martingale condition Eq(3.8) would be useless. That is, even if Eq(3.8) holds, $\hat{V},\hat{q}$ can not provide any information about current policy without Bellman condition.
> (ii) Secodnly, give the importance of Bellman condition, however, Eq(3.9) is almost impossible to impose in q-learning, and consequently the estimated $\hat{V},\hat{q}$ cannot provide guidance on policy update in most cases. Therefore, q-learning may not even get improved return in the training progress, especially for inherently difficult learning task, e.g., when $h$ is small, as shown in Fig. 2(g,k).
> (iii) Finally, when the learning task is relatively simple ($h$ is large), q-learning becomes more likely to satisfy Eq(3.9) and the martingale characterization, therefore has the chance to see improved return. Nevertheless, it is still possible that Eq(3.9) is not satisfied and q-learning degrades. Hence q-learning exhibit large varaince as shown in Fig.2(c), compared to Fig.2(g,k), while getting higher return in the average sense.
>
> [1] Treven, L., Hübotter, J., Sukhija, B., Dorfler, F., & Krause, A. (2023). Efficient exploration in continuous-time model-based reinforcement learning. Advances in Neural Information Processing Systems, 36, 42119-42147.
>
> [2] Treven, Lenart, et al. "When to sense and control? a time-adaptive approach for continuous-time RL." Advances in Neural Information Processing Systems 37 (2024): 63654-63685.
>
> [3] Zhao, R., Yu, Y., Zhu, A. Y., Yang, C., & Zhou, D. Sample and Computationally Efficient Continuous-Time Reinforcement Learning with General Function Approximation. In The 41st Conference on Uncertainty in Artificial Intelligence.
>
> [4] Zhao, R., Yu, Y., Wang, R., Huang, C., & Zhou, D. (2025). Instance-Dependent Continuous-Time Reinforcement Learning via Maximum Likelihood Estimation. arXiv preprint arXiv:2508.02103.
>
> [5] Yildiz, C., Heinonen, M., & Lähdesmäki, H. (2021, July). Continuous-time model-based reinforcement learning. In International Conference on Machine Learning (pp. 12009-12018). PMLR.

---

### Official Review · Reviewer_a8Pk · 2025-11-01

**Soundness:** 3
**Presentation:** 3
**Contribution:** 3
**Rating:** 4
**Confidence:** 4

**Summary:**

The paper formulates RL directly in continuous time (finite‑horizon SDE dynamics) and derives a DPG (Deterministic Policy Gradient) formula using an *advantage rate* (A_\phi(t,x,a)\coloneqq \mathcal{L}[V_\phi](t,x,a)+r(t,x,a)), where (\mathcal{L}) is the generator of the diffusion. The main result (Thm. 3.1) expresses (\partial_\phi V_\phi) as an expectation over (\partial_\phi \mu_\phi^\top \partial_a A_\phi), i.e., a clean continuous‑time analogue of discrete‑time DPG.  Thm. 3.2 characterizes both the value (V_\phi) and the advantage rate (A_\phi) via a *martingale orthogonality* condition. Practically, the authors enforce this by minimizing a “martingale loss” with test functions chosen as the networks’ parameter gradients and by reparameterizing the critic to satisfy (q_\psi(t,x,\mu_\phi(t,x))=0) (Eq. 4.1). This yields an implementable objective without sampling over actions.  Building on this the paper proposes **Continuous‑Time DDPG (CT‑DDPG)** (Alg. 1). Key ingredients: (i) a *multi‑step* TD objective over a short fixed window (Lh=\delta), (ii) a target value network, replay buffer, and Gaussian exploration noise, and (iii) a discrete‑time implementation robust as (h\to0). The “martingale loss” is the core critic objective (Eq. 4.2).

**Strengths:**

The DPG formula (Thm. 3.1) and the martingale identification (Thm. 3.2) are clean and connect continuous‑time stochastic control with practical deep RL estimators. The reparameterization (q_\psi=\bar q_\psi-\bar q_\psi(\cdot,\mu_\phi)) enforces the Bellman side condition  (Eq. 4.1).

The variance analysis (Props. 4.1–4.2) explains a widely observed pathology: one‑step TD becomes unusable as (h) shrinks, whereas multi‑step with fixed physical window (\delta) remains well‑behaved.

CT‑DDPG combines these ideas into a simple recipe (Alg. 1, p. 7–8): multi‑step critic, target network, replay, plus an actor trained by the estimated (q). The experiments consistently show improved stability under smaller (h) and higher noise, the regimes where discrete‑time baselines degrade (Fig. 1).

**Weaknesses:**

The paper proves bounded‑variance gradients and non‑vanishing signal for the critic objective (Props. 4.1–4.2), but it does **not** provide convergence or stability guarantees for the *actor–critic* recursion as a whole (with function approximation, target nets, replay, and exploration noise). The term “provable stability and efficiency” (Sec. 4) overstates what is established.

Clarifying the exact scope of guarantees would help. Discrete‑time baselines use standard one‑step updates; the paper does not include multi‑step or TD(λ) versions of DDPG/SAC that might mitigate the very issue analyzed here.

Among continuous‑time methods, the comparison is to a stochastic‑policy q‑learning implementation; there is no head‑to‑head with continuous‑time PPO/TRPO‑style approaches cited in Sec. A (related work). A stronger baseline suite would isolate the benefits of deterministic policies from those of *multi‑step TD*.

 Experiments inject i.i.d. Gaussian generalized forces at the MuJoCo level.  While sensible, this perturbation may not match the SDE regularity assumptions used in analysis (e.g., uniform ellipticity in Prop. 4.1’s setup). Some discussion of this modeling gap would be needed.

Theorems assume (V_\phi\in C^{1,2}) and smooth dependence on parameters/actions (Assumptions 1–2, Sec. 3). In practice, ReLU networks violate (C^1) smoothness; the paper informally argues Lipschitzness suffices (Asp. 1), but it is unclear whether subgradient analogues cover the results used in Thm. 3.1/3.2. A brief justification or reference would strengthen the story.

Identification of (A_\phi) holds in a *neighborhood* (O_{\mu_\phi(t,x)}) of the policy action (Eq. 3.7). This entails a coverage condition on exploratory actions and suggests potential brittleness far from the current policy. The algorithm’s off‑policy robustness as exploration grows is not directly tested.

 CT‑DDPG hinges on window length (\delta=Lh) and the choice of test functions in the martingale loss. Ablations over (L), (\delta), the terminal‑value penalty weight, and the set of test functions would clarify how performance depends on these choices. (Alg. 1; Eq. 4.2.)

 Tasks are standard but mid‑scale; there’s no Ant/Humanoid evaluation. The return curves (Figs. 1–2) demonstrate trends, but sample‑efficiency comparisons in terms of environment interactions (not just episodes) would make claims about efficiency more concrete.

**Questions:**

Add *n‑step* or TD(λ) DDPG/SAC and TD3; test whether multi‑step alone narrows the gap to CT‑DDPG when (h) is small.

Show learning speed/variance vs. (\delta) at fixed (h) and vs. (L) at different (h).

 Probe how performance changes with exploration noise magnitude; does the martingale identification break when actions stray far from (O_{\mu_\phi})? (Eq. 3.7.)

Beyond (\zeta_t=\partial_\theta V_\theta) or (\partial_\psi q_\psi), try basis functions in (t) and (x) to assess bias/variance of the martingale loss. (Sec. 4.1.)

Include continuous‑time TRPO/PPO‑style methods from the related‑work section and report stability under small (h).

Compare the current MuJoCo force‑noise to process‑noise injected at the state‑equation level to align more closely with the SDE assumptions. (Sec. 5.)

---

> ### Author Response · Authors · 2025-11-26
>
> Thanks for your careful review and constructive feedback! We address your concerns as follows.
>
> **W1**: The term “provable stability and efficiency” (Sec. 4) overstates what is established.
>
> **A1**: Thanks for your comments and we have modified the claim in the updated version: "we propose CT-DDPG, a novel and practical actor-critic algorithm with stability and efficiency in continuous-time environments." This paper takes the first step towards learning deterministic policies in the continuous time regime. Further convergence or stability guarantees of the proposed algorithm is left as future work.
>
> **W2**: The paper does not include multi‑step TD versions of DDPG/SAC.
>
> **A2**: Thank you for this comment. In the updated version, we have added the multi-step TD to evaluate DDPG, as shown  in Fig.1. These new  results show that "Incorporating DDPG with multi-step TD can partially alleviate this issue; however, the method still exhibits slow convergence due to the persistent bias in the TD objective induced by off-policy sampling". We provide detailed anlalysis in Appendix C.
>
> **W3**: There is no comparison with continuous‑time PPO/TRPO‑style approaches.
>
> **A3**: We would like to emphasize that our main focus is on **model-free, off-policy** settings. PPO/TRPO-style approaches are inherenly **on-policy** algorithms and requires continuously on-policy exploration which comes at the cost of substantially lower sample efficiency compared to off-policy algorithms like DDPG/SAC. In addition, (Zhao et al., 2023) only conducted experiments on simple LQ control tasks and did not release their code implementation, and thus reproducibility for fair comparison is challenging.
>
> **W4**: Experiments inject i.i.d. Gaussian generalized forces at the MuJoCo level, which may not match the uniform ellipticity in Prop 4.1.
>
> **A4**: We remark that the uniform ellipticity in Prop 4.1 is NOT necessary, and mostly for ease of exposition. In the updated version of Prop 4.1, we only "Assume $\partial_\theta V_\theta, \|\partial_xV_\theta\|_{\sigma\sigma^\top}$ are not identically zero", which is a  mild assumption and easy to satisfy even when the uniform ellipticity does not hold. In kinematic dynamics such as MuJoCo, the state typically contains position and velocity, where the position satisfies the ODE driven by velocity, and the velocity satisfies SDE driven by external (random) force. The updated Prop 4.1 is still valid in this practical scenario.
>
> **W5**: It is unclear whether subgradient analogues of $V^\phi$ cover the results used in Thm. 3.1/3.2.
>
> **A5**: The assumptions such as (V_\phi\in C^{1,2}) are common in continuous-time RL literatures (Jia and Zhou, 2022a,2023b;2023), and SDE regularity literatures [1]. These are necessary for mathematical rigor. In practice, although the ReLU activation is not smooth, there are many smooth variants of ReLU such as Softplus, which have similar performance with ReLU. Therefore Thm 3.1/3.2 are still highly consistent with practice.
>
> **W6**: Identification of $A^\phi$ entails a coverage condition on exploratory actions in the neighborhood of $O_{\mu_\phi(t,x)}$ and suggests potential brittleness far from the current policy. The algorithm’s off‑policy robustness as exploration grows is not directly tested.
>
> **A6**: We focus on **off-policy** setting where the exploration policy is typical implemented by injecting a (Gaussian) noise of a moderate scale to the current deterministic policy. *Theoretically*, the exploration policy is fully supported over the whole action space and thus it can indeed cover any neighborhood of $\mu_\phi(t,x)$. *Practically*, the TD objective is estimated via finitely many trajectories drawn from replay buffer, as typically done in off-policy discrete algorithms such as  DDPG and SAC, where the exploration policy is also updating along with the current policy through the training process. This is different from the **offline RL** setting, where all the trajectories are collected from a behavior policy irrelevant to current policy. To conclude, introducing large-scale noise to test "robustness as exploration grows" is practically adversarial and counter-productive, therefore not a practical approach for our setting.
>
> **W7**: Ablations over window length $\delta,L,h$.
>
> **A7**: We conduct ablation studies in Fig.4 in the updated version. "The performance of choosing $L\propto 1/h$ remains robust across a wide range of $h$ and consistently outperforms the DAU baseline (i.e., $L=1$), validating our theoretical results in Prop 4.2. Note that all choices of $L$ deteriorate when $h$ becomes extremely small, as the resulting state-transition noise dominates the dynamics.

---

> ### Author Response · Authors · 2025-11-26
>
> **W8**: The x-axis of return curves (Figs. 1–2) should be environment interactions (not just episodes) in order to make claims about efficiency more concrete. The results demonstrate that "The performance of choosing $L\propto 1/h$ remains robust across a wide range of $h$ and consistently outperforms the DAU baseline (i.e., $L=1$), validating our theoretical results in Prop 4.2. Note that all choices of $L$ deteriorate when $h$ becomes extremely small, as the resulting state-transition noise dominates the dynamics.
>
> **A8**: In each figure, the interaction frequencies of different algorithms are identical and fully determined by the step size $h$. Therefore the number of episodes is equivalent to the number of interactions. We use episode as the x-axis to present results obtained under the same dynamics but different step sizes in a cleaner and more interpretable manner.
>
> [1] Beck, Christian, Martin Hutzenthaler, and Arnulf Jentzen. "On nonlinear Feynman–Kac formulas for viscosity solutions of semilinear parabolic partial differential equations." Stochastics and Dynamics 21.08 (2021): 2150048.

---

### Official Review · Reviewer_8UWq · 2025-11-01

**Soundness:** 3
**Presentation:** 3
**Contribution:** 3
**Rating:** 6
**Confidence:** 4

**Summary:**

This paper proposes CT-DDPG, an discrete RL algorithm in the continuous RL paradigm. The main idea is to use the martingale orthogonality characterization. Experiments have been conducted on Pendulum-v1, HalfCheetah-v5, Hopper-v5, and Walker2d-v5.

**Strengths:**

The paper is well-written, and I mostly enjoyed reading it. The theoretical results are solid. The numerical experiments also support the proposed approach.

**Weaknesses:**

Below are a few comments:

(1) The introductory part seems to be too long, e.g., Thm 3.1 and 3.2 are basically known from the literature. It isn't until Section 4 (p.5) that new results come. The authors may think of a bit reorganization.

(2) The idea of applying Robbins-Monroe in continuous RL was also explored in Regret of exploratory policy improvement and q-learning, arXiv:2411.01302, Tang and Zhao -- but using the martingale loss approach. This paper considers a different martingale orthogonality approach. See also On Optimal Tracking Portfolio in Incomplete Markets: The Reinforcement Learning Approach, SICON, Bo, Huang and Yu for a similar analysis in a specific setting.

(3) Proposition 4.2 (and also 4.1): the authors proved the variance blow-up/no blow-up. However, there is no analysis on the proposed algorithm. What is (roughly) the value function gap in terms of h? Also the proposed algorithm uses the number of steps of order 1/h -- which is exploding. The authors may provide some explanations.

(4) Is there any insight how to choose h in practical problems?

**Questions:**

See weaknesses.

---

> ### Author Response · Authors · 2025-11-26
>
> Thanks for your careful review and constructive feedback! We address your concerns as follows.
>
> **W1**: The introductory part seems to be too long, e.g., Thm 3.1 and 3.2 are basically known from the literature.
>
> **A1**: We'd like to emphasize that **Thm 3.2 is fundamentally different from the martingale results Thm 7 in (Jia and Zhou, 2023) and is one of the core contributions of this paper**. Thm 7 in (Jia and Zhou, 2023) can only characterize the advantage-rate function for ***fully supported*** stochastic policy and imposes Bellman condition $\mathbb{E}_{a\sim \pi(a|t,x)}[\hat q(t,x,a)-\gamma\log\pi(a|t,x)] = 0$, which is impossible to satisfy for general neural networks. In contrast, our Thm 3.2 provides a martingale characerization for deterministic policy, which can be readily plugged in off-policy settings. In addition, the Bellman condition in Eq(3.6) can be easily satisfied via a reparameterization trick Eq(4.1), leading to the fundamental benefits of learning deterministic policy compared to (Jia and Zhou, 2023).
> Moreover, Thm 3.1 is technically novel since it removes most stringent conditions in existing works as discussed in line 163-165.
>
>
> **W2**: Martingale orthogonality approach.
>
> **A2**: Thanks for mentioning the reference and we have added the citation in Sec 4.1 in the updated version.
>
> **W3**: There is no analysis on the proposed algorithm. What is (roughly) the value function gap in terms of h? Also the proposed algorithm uses the number of steps of order 1/h.
>
> **A3**: (i) We establish Prop 4.1, 4.2 in order to provide the theoretical insights of the failure of conventional discrete-time RL algorithms using one-step TD in continuous-time regime, highlighting the critical role of the multi-step TD. We agree that analyzing explicitly the discretization error in continuous-time MDP is beyond the scope of this paper. Also the theoretical anlalysis of CT-DDPG such as its convergence is very interesting and worth studying in the future (ii) We further conduct ablation studies for the choice of $L$ in line 462-475. It turns out that the choice of  $L\propto 1/h$ provides robust results. We would like to point out that when $h$ is too small,  the state transition becomes too noisy, and the algorithm may degrade regardless the choice of $L$.
>
> **W4**:  Is there any insight how to choose h in practical problems?
>
> **A4**: In our setting, $h$ is assumed to be specified by the underlying system and therefore treated as a prior. In the cases where $h$ can indeed be selected, one should balance the trade-off between control accuracy and learning difficulty. As discussed above, $h$ being too small can make the state-transition dynamics extremely noisy, which can potentially cause continuous-time algorithms to degrade in practice.

---

### Official Review · Reviewer_s7XK · 2025-11-10

**Soundness:** 3
**Presentation:** 3
**Contribution:** 3
**Rating:** 6
**Confidence:** 1

**Summary:**

The paper studies the setting of policy gradient in continuous time and state-action spaces. The authors essentially rely on a martingale characterization to implement PG in a continuous setting. The PG theorem derived is analogous to the discrete counterpart with the advantage rate function taking the place of the advantage function. The authors list a bunch of regularity conditions to invoke the existence of Bellman equation and so on. The policies are deterministic mappings from state to action. The authors then argue that TD(0) is fundamentally incompatible in the continuous setting and propose a multi step TD approach in its place. The proposed algorithm is experimentally validated.

**Strengths:**

The paper considers continuous RL which is technically challenging and not that well studied.

Identifies the fundamental failure of TD(0) in this setting.

Considers deterministic policies which reduces computational cost that is incurred due to sampling.

**Weaknesses:**

Considers the finite horizon setting only. What are some challenges when it comes to extending the theory to the infinite horizon settings?

The regularity assumptions _can_ be strong and may not always be met.

**Questions:**

What is the form of Q and V in line 182?

How is the neighbourhood O determined in line 196?

---

> ### Author Response · Authors · 2025-11-26
>
> Thanks for your careful review and constructive feedback! We address your concerns as follows.
>
> **W1**: Considers the finite horizon setting only. What are some challenges when it comes to extending the theory to the infinite horizon settings?
>
> **A1**: Thanks for poinitng this out. It is true that our paper focuses on the finite horizon setting. Nevertheless, the extension to the infinite time horizon is technically similar, and hence omitted to avoid repetition. We have added this comment in the updated version.
>
> **W2**: The regularity assumptions can be strong and may not always be met.
>
> **A2**: The regularity assumptions including Asp 1,2 are commonly adopted in the SDE literatures (for instance in Jia and Zhou, 2022a;2022b;2023) in order to ensure the existence and regularity of the state dynamics. These assumptions are necessary for the well defindness of the stochastic system, from which the martingale characterization can be established. In practice,  the efficiency of proposed CT-DDPG algorithm goes beyond this theorectical setting, as shown in our experiments.
>
> **Q1**: What is the form of Q and V in line 182?
>
> **A3**: Q and V satisfy the classic definition of Q-function and value function in discrete-time RL, i.e.,  $Q^{\Delta t, \phi}(t_i,x,a):=\mathbb{E}[\sum_{j=i}^{N-1}r(t_j,x_{t_j},a_{t_j})+g(x_T^{\Delta t,\phi})\mid x_{t_i}=x,a_{t_i}=a]$ and $V^{\Delta t, \phi}(t_i,x):=Q(t_i,x,\mu_\phi(t_i,x))$.
>
> **Q2**: How is the neighbourhood O determined in line 196?
>
> **A4**: Thm 3.2 only requires the existence of a neighborhood around the deterministic policy where the martingale condition holds. In practice, since the exploration policy is typically defined by adding some (Gaussian) noise to the deterministic policy, the exploration policy is fully supported over the action space. In this case, the neighbourhood can be simply defined as the whole action space.

---

### Meta-Review · Area_Chair_dBF9 · 2025-12-26

**Summary:**

Reviewers generally agreed that the paper addresses a technically challenging and important problem, and that the use of martingale characterizations to derive a deterministic policy gradient formulation is mathematically sound and well-motivated. The work was viewed as careful, principled, and supported by nontrivial theory and experiments. However, several recurring concerns tempered overall enthusiasm, leading to a more cautious assessment: 1) incremental novelty relative to prior continuous rl work; 2) lack of convergence or stability guarantee for the actor-critic algorithm.

**Reviewer Concerns:**

1. Novelty relative to prior work: This concern was addressed
2. No full convergence results: not addressed
3. missing multi-step td basedline: addressed
4. choice of h and delta: addressed
5. baseline of continous RL: not fully addressed

**Reviewer Scores:**

Reviewer s7XK: 6->6
Reviewer 8UWq: 6->6
Reviewer a8Pk: 4->4
Reviewer C2As: 2->2

---

### Decision · Program_Chairs · 2026-01-26

Reject